# Towards Automated Circuit Discovery
# for Mechanistic Interpretability

**Arthur Conmy**[*]      **Augustine N. Mavor-Parker**[*]   **Aengus Lynch**[*]   **Stefan Heimersheim**
Independent                      UCL                             UCL              University of Cambridge

**Adrià Garriga-Alonso**[*]
FAR AI

## Abstract

Through considerable effort and intuition, several recent works have reverse-engineered nontrivial behaviors of transformer models. This paper systematizes the mechanistic interpretability process they followed. First, researchers choose a metric and dataset that elicit the desired model behavior. Then, they apply activation patching to find which abstract neural network units are involved in the behavior. By varying the dataset, metric, and units under investigation, researchers can understand the functionality of each component.

We automate one of the process' steps: finding the connections between the abstract neural network units that form a circuit. We propose several algorithms and reproduce previous interpretability results to validate them. For example, the ACDC algorithm rediscovered 5/5 of the component types in a circuit in GPT-2 Small that computes the Greater-Than operation. ACDC selected 68 of the 32,000 edges in GPT-2 Small, all of which were manually found by previous work. Our code is available at https://github.com/ArthurConmy/Automatic-Circuit-Discovery.

## 1   Introduction

Rapid progress in transformer language modelling (Vaswani et al., 2017; Devlin et al., 2019; OpenAI, 2023, *inter alia*) has directed attention towards understanding the causes of new capabilities (Wei et al., 2022) in these models. Researchers have identified precise high-level predictors of model performance (Kaplan et al., 2020), but transformers are still widely considered 'black-boxes' (Alishahi, Chrupała, and Linzen, 2019) like almost all other neural network models (Fong and Vedaldi, 2017; Buhrmester, Münch, and Arens, 2021).[2] Interpretability research aims to demystify machine learning models, for example by explaining model outputs in terms of domain-relevant concepts (Zhang et al., 2021).

Mechanistic interpretability focuses on reverse-engineering model components into human-understandable algorithms (Olah, 2022). Much research in mechanistic interpretability views models as a computational graph (Geiger et al., 2021), and circuits are subgraphs with distinct functionality (Wang et al., 2023). The current approach to extracting circuits from neural networks relies on a lot of manual inspection by humans (Räuker et al., 2022). This is a major obstacle to scaling up mechanistic interpretability to larger models, more behaviors, and complicated behaviors composed of many sub-circuits. This work identifies a workflow for circuit research, and automates part of it by presenting several methods to extract computational graphs from neural networks.

Our main contributions are as follows. First, we systematize the common workflow prevalent in many existing mechanistic interpretability works, outlining the essential components of this process

---

[*]Work partially done at Redwood Research. Correspondence to arthurconmy@gmail.com

[2]Though this perspective is not universal (Lipton, 2016).

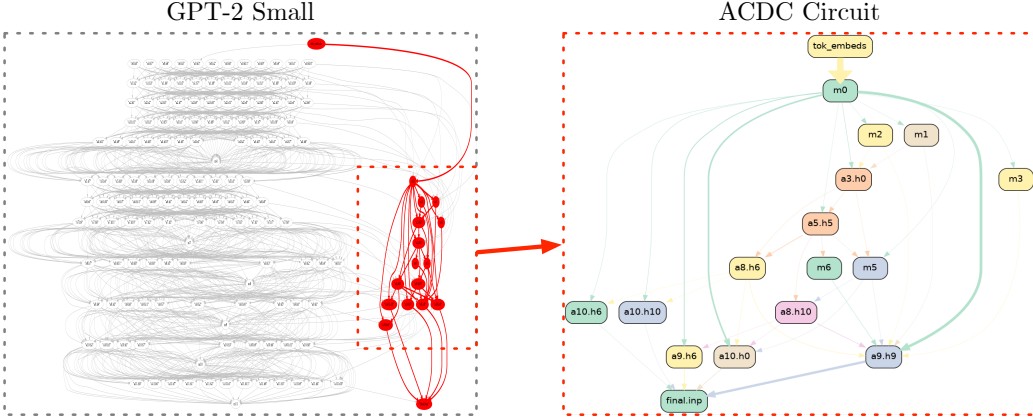

Figure 1: **Automatically discovering circuits with ACDC.** *Left:* a computational graph for GPT-2 Small, with a recovered circuit for the IOI task highlighted in red. Only edges between adjacent layers are shown. *Right:* the recovered circuit with labelled nodes. All heads recovered were identified as part of the IOI circuit by Wang et al. (2023). Edge thickness is proportional to importance.

(Section 2). One of its steps is to find a subgraph of the model which implements the behavior of interest, which is a step possible to automate. We introduce Automatic Circuit DisCovery (ACDC), a novel algorithm that follows the way in which researchers identify circuits (Section 3), and adapt Subnetwork Probing (SP; Cao, Sanh, and Rush, 2021) and Head Importance Score for Pruning (HISP; Michel, Levy, and Neubig, 2019) for the same task. Finally, we introduce quantitative metrics to evaluate the success of circuit extraction algorithms (Sections 4 and 4.2). We present a detailed ablation study of design choices in Appendix E and qualitative studies in Appendices F, G, H, I and J.

## 2 The Mechanistic Interpretability Workflow

Mechanistic interpretability attempts to explain and predict neural network behaviors by understanding the underlying algorithms implemented by models. In the related work section we discuss the mechanistic interpretability field and its relationship to 'circuits' research (Section 5). Neural network behaviors are implemented by algorithms within the model's computational graph, and prior work has identified subgraphs (*circuits*, following Wang et al. (2023)'s definition) that capture the majority of particular behaviors. In this section, we describe a workflow that several prior works have followed that has been fruitful for finding circuits in models.

As a concrete example of an approach taken to finding a circuit, Hanna, Liu, and Variengien (2023) prompt GPT-2 Small with a dataset of sentences like "The war lasted from 1517 to 15". GPT-2 Small completes this sentence with "18" or "19" or any larger two digit number, but not with any two digit number that is at most "17" (from here, we refer to prompt completions like this as the "Greater-Than" task). This behavior can be measured by the difference in probability the model places on a completion "18" or "19" or larger and the probability the model places on a completion "17" or smaller. Note that we use the term 'dataset' to refer to a collection of prompts that elicit some behavior in a model: we do not train models on these examples, as in this paper we focus on post-hoc interpretability.

The researchers then create a *corrupted dataset* of sentences that do not have any bias against particular two digit completions (the '01-dataset' (Hanna, Liu, and Variengien, 2023)). The researchers attribute the greater-than operation to late layer MLPs and then find earlier components that identify the numerical values of years, including attention heads in the model. Finally, Hanna, Liu, and Variengien (2023) interpret the role of each set of components. For example, they identify early model components that respond to the "17" token, and later model components that boost the importance of logits for years greater than 17.

There are equivalent steps taken in a growing number of additional works (Heimersheim and Janiak, 2023, the "Docstring" task; Goldowsky-Dill et al., 2023, the "Induction" task; Wang et al., 2023,

the "IOI" task), described in brief in Table 1 and in detail in Appendices F, H and J. We identify the workflow that eventually finds a circuit as following three steps. Researchers:

1. Observe a behavior (or task[3]) that a neural network displays, create a dataset that reproduces the behavior in question, and choose a metric to measure the extent to which the model performs the task.

2. Define the scope of the interpretation, i.e. decide to what level of granularity (e.g. attention heads and MLP layers, individual neurons, whether these are split by token position) at which one wants to analyze the network. This results in a computational graph of interconnected model units.

3. Perform an extensive and iterative series of patching experiments with the goal of removing as many unnecessary components and connections from the model as possible.

Researchers repeat the previous three steps with a slightly different dataset or granularity, until they are satisfied with the explanation of the circuit components.

This work (ACDC) presents a tool to fully automate Step 3. Before we dive into the details of ACDC, we expand on what Steps 1-3 involve, and review examples from previous work that we use to evaluate ACDC.

## 2.1    Step 1: Select a behavior, dataset, and metric

The first step of the general mechanistic interpretability workflow is to choose a neural network behavior to analyze. Most commonly researchers choose a clearly defined behavior to isolate only the algorithm for one particular task, and curate a dataset which elicits the behavior from the model. Choosing a clearly defined behavior means that the circuit will be easier to interpret than a mix of circuits corresponding to a vague behavior. Some prior work has reverse-engineered the algorithm behind a small model's behavior on all inputs in its training distribution (Nanda et al., 2023; Chughtai, Chan, and Nanda, 2023), though for language models this is currently intractable, hence the focus on individual tasks.

We identified a list of interesting behaviors that we used to test our method, summarized in Table 1. These include previously analyzed transformer models (1 and 3 on GPT-2 Small, 2 and 6 on smaller language transformers) where researchers followed a workflow similar to the one we described above. Tasks 4 and 5 involve the full behavior of tiny transformers that implement a known algorithm, compiled with `tracr` (Lindner et al., 2023). For each task, we mention the metric used in previous work to measure the extent to which the model performs the task on the corresponding dataset.

## 2.2    Step 2: Divide the neural network into a graph of smaller units

To find circuits for the behavior of interest, one must represent the internals of the model as a computational directed acyclic graph (DAG, e.g. Figure 2a). Current work chooses the abstraction level of the computational graph depending on the level of detail of their explanations of model behavior. For example, at a coarse level, computational graphs can represent interactions between attention heads and MLPs. At a more granular level they could include separate query, key and value activations, the interactions between individual neurons (see Appendix I), or have a node for each token position (Wang et al., 2023).

Node connectivity has to be faithful to the model's computation, but that does not fully specify its definition. For example, following Elhage et al. (2021), many works consider the connections between model components in non-adjacent layers due to the additivity of the residual stream, even though these are computed with dynamic programming in the actual model implementation. Connectivity defines what is considered a direct or a mediated interaction (Pearl, 2009; Vig et al., 2020). See for example Figure 2a, where component B has both a direct effect on the output node O and an indirect effect on the output through component A.

---

[3]Section 3 formally defines "task". We use "behavior" and "task" interchangeably.

| Task | Example Prompt | Output | Metric |
|---|---|---|---|
| 1: IOI (Appendix F.2) | "When John and Mary went to the store, Mary gave a bottle of milk to" | "_John" | Logit difference |
| 2: Docstring (Appendix H.1) | `def f(self, files, obj, state, size, shape, option):`
`    """document string example`

`    :param state: performance analysis`
`    :param size: pattern design`
`    :param` | "_shape" | Logit difference |
| 3: Greater-Than (Appendix G) | "The war lasted from 1517 to 15" | "18" or "19" or . . . or "99" | Probability difference |
| 4: tracr-xproportion (Appendix I.1) | `["a", "x", "b", "x"]` | `[0, 0.5, 0.33, 0.5]` | Mean Squared Error |
| 5: tracr-reverse (Appendix I.2) | `[0, 3, 2, 1]` | `[1, 2, 3, 0]` | Mean Squared Error |
| 6: Induction (Section 4.2) | "Vernon *Durs*ley and Petunia *Durs*" | "ley" | Negative log-probability |

Table 1: Five behaviors for which we have an end-to-end circuit from previous mechanistic interpretability work, plus Induction. We automatically rediscover the circuits for behaviors 1-5 in Section 4. Tokens beginning with space have a "_" prepended for clarity.

## 2.3 Step 3: Patch model activations to isolate the relevant subgraph

With the computational DAG specified, one can search for the edges that form the circuit. We test edges for their importance by using recursive *activation patching*: i) overwrite the activation value of a node or edge with a corrupted activation, ii) run a forward pass through the model, and iii) compare the output values of the new model with the original model, using the chosen metric (Section 2.1). One typically starts at the output node, determines the important incoming edges, and then investigates all the parent nodes through these edges in the same way. It is this procedure that ACDC follows and automates in Algorithm 1.

**Patching with zeros and patching with different activations**    Activation patching methodology varies between mechanistic interpretability projects. Some projects overwrite activation values with zeros (Olsson et al., 2022; Cammarata et al., 2021), while others erase activations' informational content using the mean activation on the dataset (Wang et al., 2023). Geiger et al. (2021) prescribe *interchange interventions* instead: to overwrite a node's activation value on one data point with its value on another data point. Chan et al. (2022) justify this by arguing that both zero and mean activations take the model too far away from actually possible activation distributions. Interchange interventions have been used in more interpretability projects (Hanna, Liu, and Variengien, 2023; Heimersheim and Janiak, 2023; Wang et al., 2023), so we prefer it. However we also compare all our experiments to replacing activations with zeros (Section 4.2, Appendix E.2).

## 2.4 Explaining the circuit components

After successfully isolating a subgraph, one has found a circuit (Section 1). The researcher then can formulate and test hypotheses about the functions implemented by each node in the subgraph. There is early evidence that ACDC is helpful for making novel observations about how language models complete tasks, such as the importance of surprising token positions that help GPT-2 Small predict correctly gendered pronouns (Appendix K). In our work we focus on automating the time-consuming step 3 that precedes functional interpretation of internal model components, though we think that automating the functional interpretation of model components is an exciting further research direction.

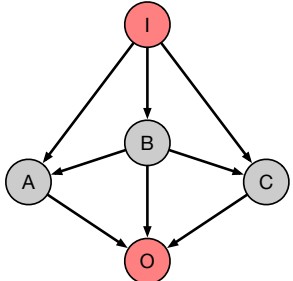 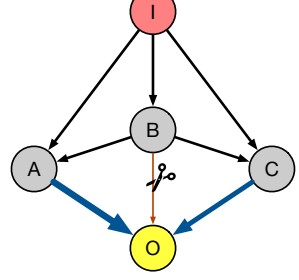 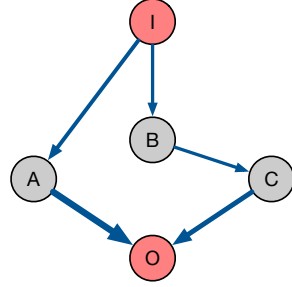

(a) Choose computational graph, task, and threshold $\tau$.

(b) At each head, prune unimportant connections.

(c) Recurse until the full circuit is recovered.

Figure 2: **How ACDC works** (Steps 2a-2c). Step 2a: a practitioner specifies a computational graph of the model, the task they want to investigate, and a threshold under which to remove connections. Step 2b: ACDC iterates over nodes in the computational graph, replacing activations of connections between a node and its children, and measuring the effect on the output metric. Connections are removed if their measured effect on the metric under corruption is below the threshold $\tau$. Step 2c: recursively apply Step 2b to the remaining nodes. The ACDC procedure returns a subgraph of the original computational graph.

# 3 Automating circuit discovery (Step 3)

This section describes algorithms to automate Step 3 of the mechanistic interpretability workflow (Section 2.3). In all three cases, we assume that the 'task' being studied is defined by a set of prompts $(x_i)_{i=1}^n$ on which the model's predictions have a noticeable pattern (see Table 1 for examples) and a set of prompts $(x_i')_{i=1}^n$ where this task is not present. We then use the activations of the models on a forward pass on the points $x_i'$ as corrupted activations (Section 2.3).

**Automatic Circuit DisCovery (ACDC).**    Informally, a run of ACDC iterates from outputs to inputs through the computational graph, starting at the output node, to build a subgraph. At every node it attempts to remove as many edges that enter this node as possible, without reducing the model's performance on a selected metric. Finally, once all nodes are iterated over, the algorithm (when successful) finds a graph that i) is far sparser than the original graph and ii) recovers good performance on the task.

To formalize the ACDC process, we let $G$ be a computational graph of the model of interest, at a desired level of granularity (Section 2.2), with nodes topologically sorted then reversed (so the nodes are sorted from output to input). Let $H \subseteq G$ be the computational subgraph that is iteratively pruned, and $\tau > 0$ a threshold that determines the sparsity of the final state of $H$.

We now define how we evaluate a subgraph $H$. **We let $H(x_i, x_i')$ be the result of the model when $x_i$ is the input to the network, but we overwrite all edges in $G$ that are not present in $H$ to their activation on $x_i'$ (the corrupted input).**[4] This defines $H(x_i, x_i')$, the output probability distribution of the subgraph under such an experiment. Finally we evaluate $H$ by computing the KL divergence $D_{KL}(G(x_i)||H(x_i, x_i'))$ between the model and the subgraph's predictions. We let $D_{KL}(G||H)$ denote the average KL divergence over a set of datapoints. Appendix C discusses alternatives to the KL divergence, and Appendix E.1 explores the consequences of optimizing the task-specific metrics from Table 1 instead.

Algorithm 1 describes ACDC. The order in which we iterate over the parents $w$ of $v$ is a hyperparameter. In our experiments the order is lexicographically from later-layer MLPs and heads to earlier-layer MLPs and heads, and from higher- to lower-indexed heads. We note that in one case in our work, the order of the parents affected experimental results (Appendix J).

**Subnetwork Probing (SP; Cao, Sanh, and Rush, 2021).**    SP learns a mask over the internal model components (such as attention heads and MLPs), using an objective that combines accuracy and

---

[4]To implement the computation of $H(x_i, x_i')$, we initially run a forward pass with the unmodified model on the input $x_i'$ and cache all activations.

**Algorithm 1:** The ACDC algorithm.

**Data:** Computational graph $G$, dataset $(x_i)_{i=1}^n$, corrupted datapoints $(x_i')_{i=1}^n$ and threshold $\tau > 0$.

**Result:** Subgraph $H \subseteq G$.

```
1  H ← G                              // Initialize H to the full computational graph
2  H ← H.reverse_topological_sort()                   // Sort H so output first
3  for v ∈ H do
4      for w parent of v do
5          H_new ← H \ {w → v}                // Temporarily remove candidate edge
6          if D_KL(G||H_new) − D_KL(G||H) < τ then
7              H ← H_new                   // Edge is unimportant, remove permanently
8          end
9      end
10 end
11 return H
```

sparsity (Louizos, Welling, and Kingma, 2018), with a regularization parameter $\lambda$. At the end of training, we round the mask to 0 or 1 for each entry, so the masked computation corresponds exactly to a subnetwork of a transformer. SP aims to retain enough information that a linear probe can still extract linguistic information from the model's hidden states. In order to use it to automate circuit discovery, we make three modifications. We i) remove the linear probe, ii) change the training metric to KL divergence as in Section 2, and iii) use the mask to interpolate between corrupted activations and clean activations (Section 3) rather than zero activations and clean activations. Appendix D.1 explains the details of these changes.

**Head Importance Score for Pruning (HISP; Michel, Levy, and Neubig, 2019).** HISP ranks the heads by importance scores (Appendix D.2) and prunes all the heads except those with the top $k$ scores. Keeping only the top $k$ heads corresponds to a subnetwork that we can compare to ACDC. We plot the ROC obtained from the full possible range of $k$. Like SP, this method only considers replacing head activations with zero activations, and therefore we once more generalize it to replace heads and other model components with corrupted activations (for details, see Appendix D.2).

## 4 Evaluating Subgraph Recovery Algorithms

To compare methods for identifying circuits, we seek empirical answers to the following questions.

- **Q1:** Does the method identify the subgraph corresponding to the underlying algorithm implemented by the neural network?
- **Q2:** Does the method avoid including components which do not participate in the elicited behavior?

We attempt to measure **Q1** and **Q2** using two kinds of imperfect metrics: some grounded in previous work (Section 4.1), and some that correspond to stand-alone properties of the model and discovered subgraph (Section 4.2).

### 4.1 Grounded in previous work: area under ROC curves

The receiver operating characteristic (ROC) curve is useful because a high true-positive rate (TPR) and a low false-positive rate (FPR) conceptually correspond to affirming **Q1** and **Q2**, respectively.

We consider *canonical* circuits taken from previous works which found an end-to-end circuit explaining behavior for tasks in Table 1. We formulate circuit discovery as a binary classification problem, where edges are classified as positive (in the circuit) or negative (not in the circuit). Appendices F, G, H, I and J describe and depict the canonical circuits for each task. Appendix E.3 considers the node classification problem instead, which is less appropriate for ACDC but more appropriate for other methods.

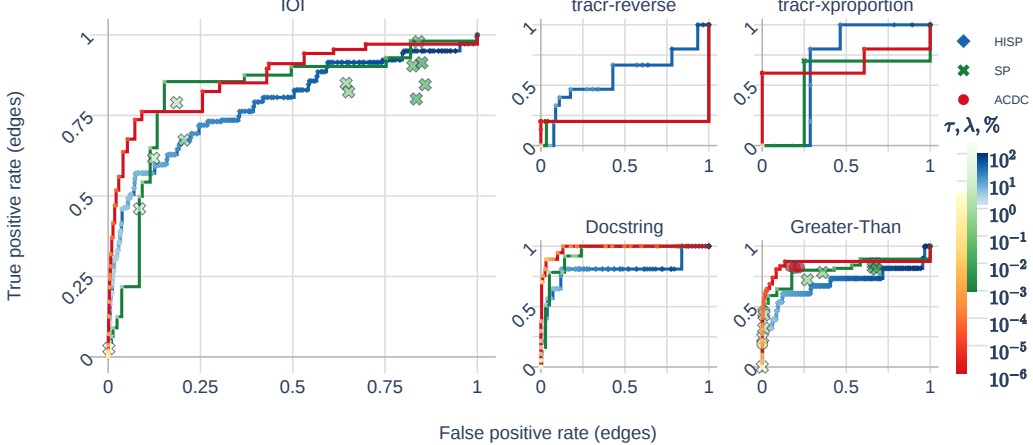

Figure 3: ROC curves of ACDC, SP and HISP identifying model components from previous work, across 5 circuits in transformers. The points on the plot are cases where SP and ACDC return subgraphs that are not on the Pareto frontier. The corresponding AUCs are in Table 2.

We sweep over a range of ACDC thresholds $\tau$, SP regularization parameters $\lambda$, or number of HISP elements pruned $k$. We plot pessimistic segments between points on the Pareto frontier of TPR and FPR, over this range of thresholds (Fawcett, 2006). ACDC and SP optimize the KL divergence for tasks where this makes sense (all but `tracr` tasks, which use the L2 distance). All methods employ activations with corrupted data. Appendix C describes and Appendix E experiments with different design choices for the metric and activation patching methodology.

Figure 3 shows the results of studying how well existing methods recover circuits in transformers. We find that i) methods are very sensitive to the corrupted distribution, ii) ACDC has competitive performance (as measured by AUC) with gradient-descent based methods iii) ACDC is not robust, and it fails at some settings.

Several of the tasks appeared to require specific distributions and metrics for the areas under the curves to be large. For example, ACDC achieved poor performance on both `tracr` tasks in Fig. 3, but the circuit was perfectly recovered by ACDC at any threshold $\tau > 0$ when patching activations with zeros (Appendix I). Furthermore, ACDC achieves a greater AUC on the IOI and Greater-Than and tracr-reverse tasks than both of the other methods, and hence overall is the optimal algorithm. As an example of the variable performance of circuit recovery algorithms, on the Docstring task we achieve the high perfomance when using the ACDC algorithm with the docstring metric (Appendix H). However in other tasks such as the IOI task, ACDC performance was worse when optimizing for logit difference.

Further research in automated interpretability will likely yield further improvements to the FPR and TPR of circuit discovery. We outline limitations with all current methods, but also gesture at likely fundamental limitations of the false positive and true positive measures. A limitation with all existing methods is that they optimize a single metric. This means they systematically miss internal model components such as the "negative" components found in previous work (IOI, Docstring) that are actively harmful for performance. The IOI recovery runs were not able to recover negative heads when optimizing for logit difference. Even when optimizing for low KL divergence, the negative components were only recovered when very small thresholds were used (Figure 15).

Additionally, a more fundamental limitation to measuring the false and true positive rates of circuit recovery methods is that the ground-truth circuits are reported by practitioners and are likely to have included extraneous edges and miss more important edges. The language model circuits studied in our work (Appendices F-H) involve a large number of edges (1041 in the case of IOI) and the full models contain more than an order of magnitude more edges. Since these interpretability works are carried out by humans who often report limitations of their understanding, our 'ground-truth' is not 100% reliable, limiting the strength of the conclusions that can be drawn from the experiments in this section.

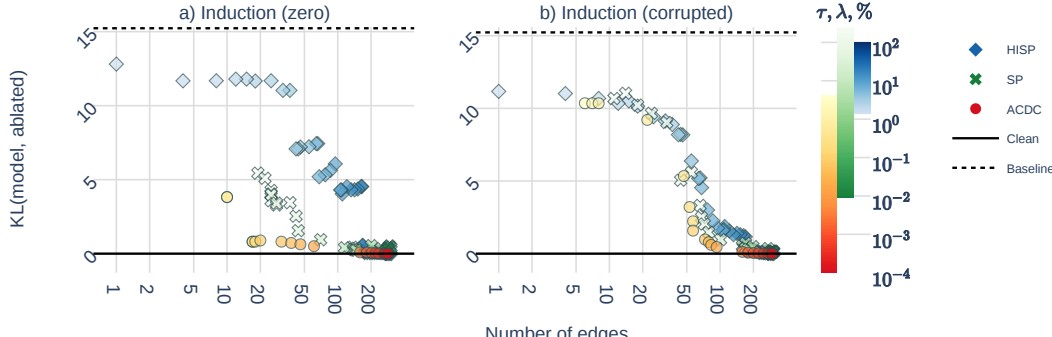

Figure 4: Comparison of ACDC and SP with both zero-input activations (left) and corrupted activations (right). We plot the KL Divergence on a held-out test set against the number of edges of each hypothesized circuit. Lower KL divergence and fewer edges correspond to better subgraphs. Darker points include more edges in the hypothesis: they use smaller ACDC $\tau$, smaller SP regularization $\lambda$ or a higher percentage of nodes in HISP.

## 4.2 Stand-alone circuit properties with a test metric

This section evaluates the algorithms by studying the induction task. We measure the KL Divergence of the circuits recovered with the three methods to the original model. This is an indirect measure of **Q1**, with the advantage of not relying on the completeness or correctness of previous works. As an indicator of **Q2**, we also measure the number of edges that a hypothesized circuit contains. A circuit with fewer edges which still obtains a low KL Divergence is less likely to contain components that do not participate in the behavior. In Appendix L we also introduce and explain experiments on **reset networks** that provide more evidence for **Q2**.

Our mainline experimental setup is to run the circuit recovery algorithms as described in Algorithm 1 and Section 3 and then measure the KL Divergence for these circuits on the induction task (Appendix J). In brief, ACDC performs better that the other methods under these experimental conditions with both corrupted and zero activations. For example, the left-hand side of Figure 4 shows that, above 20 edges, ACDC starts having a slight advantage over other methods in terms of behavior recovered per number of edges as all points on the Pareto-frontier with at least this many edges are generated from ACDC runs. Appendix E describes many further experiments with variations on setup to provide a more complete picture of the performance of the circuit recovery algorithms. For example, when we measure the loss (the task-specific induction metric; Table 1) of subgraphs recovered by optimizing KL Divergence, we find very similar qualitative graphs to Figure 4.

In Appendix L we see that the KL divergence that all methods achieve is significantly lower for the trained networks, indicating that all the methods get signal from the neural network's ability to perform induction (Figure 4). HISP and SP with zero activations, and to some extent SP with corrupted activations are also able to optimize the reset network. This suggests that these methods are somewhat more prone to finding circuits that don't exist (i.e. evidence against **Q2**).

## 5 Related work

**Mechanistic interpretability** encompasses understanding features learnt by machine learning models (Olah, Mordvintsev, and Schubert, 2017; Elhage et al., 2022), mathematical frameworks for understanding machine learning architetures (Elhage et al., 2021) and efforts to find *circuits* in models (Nanda et al., 2023; Cammarata et al., 2021; Chughtai, Chan, and Nanda, 2023; Wang et al., 2023). The higher standard of a mechanistic understanding of a model has already had applications to designing better architectures (Fu et al., 2023), though the speculative goal of mechanistic interpretability is to understand the behavior of whole models, perhaps through describing all their circuits and how they compose. Little work has been done to automate interpretability besides Bills et al. (2023) who use language models to label neurons in language models.

**Neural network pruning** masks the weights of neural networks to make their connectivity more sparse (LeCun, Denker, and Solla, 1989). In contrast to our aims, the pruning literature is typically concerned with compressing neural networks for faster inference or to reduce storage requirements (Wang, Wohlwend, and Lei, 2020; Kurtic et al., 2022). Early work (Hassibi and Stork, 1992) hoped pruning would lead to more interpretable networks, but progress towards interpretability via pruning is limited (Grover, Gawri, and Manku, 2022).

Pruning techniques may learn masks from data, which is a special case of more generally using gradient information. Masks can also be learned from data, with an objective function that balances model performance and network sparsity (Louizos, Welling, and Kingma, 2018; Wang, Wohlwend, and Lei, 2020; Cao, Sanh, and Rush, 2021). This is a useful comparison to ACDC as learnable masks do not change the weights of our model after pruning (Frantar and Alistarh, 2023). Examples of gradient information being used more generally includes Michel, Levy, and Neubig (2019) who decide which heads should be pruned by using the absolute value of their gradients, while "movement pruning" (Sanh, Wolf, and Rush, 2020) removes parameters that have high velocity to a low magnitude. ACDC is different from pruning and other compression techniques (Zhu et al., 2023) since i) the compressed networks we find are reflective of the circuits that model's use to compute outputs to certain tasks (Section 4) and ii) our goal is not to speed up forward passes, and generally our techniques slow forwards passes.

**Causal interpretation**. Much prior research on understanding language models has drawn inspiration from causal inference (Pearl, 2009), leading to the development of frameworks that provide causal explanations for model outputs (Pearl, 2009; Feder et al., 2021; Geiger et al., 2021; Wu et al., 2022; Kaddour et al., 2022). Other work (Vig et al., 2020) discusses the difference between indirect effects and direct effects inside language models, and experiments on removing subsets of these heads using heads' direct effects as proxies for the overall contribution of these heads. Goldowsky-Dill et al. (2023) introduce 'path patching' to analyze the effects of different subsets of edges in computational graphs of models. In parallel to our work, Wu et al. (2023) develop a method to automatically test whether neural networks implement certain algorithms with causal testing. Our work is focused on finding rather than verifying an outline of an algorithm implemented by a model.

**Computational subgraphs for interpretability.** Training dynamics in residual models can be explained by shallow paths through the computational graph (Veit, Wilber, and Belongie, 2016). MLP layers can be modelled as memory that is able to represent certain properties of the network inputs (Geva et al., 2021). Residual transformer models have been modelled as the sum of all different paths through the network (Elhage et al., 2021). Later work has used insights from looking at subgraphs of models in order to edit models' behaviors (Bau et al., 2020; Meng et al., 2022) and test interpretability hypotheses (Chan et al., 2022).

## 6 Conclusion

We have identified a common workflow for mechanistic interpretability. First, pin down a behavior using a metric and data set. Second, conduct activation patching experiments to understand which abstract units (e.g. transformer heads) are involved in the behavior. Third, iterate the previous steps with variations of the behavior under study, until the model's algorithm is understood.

The main proposed algorithm, ACDC, systematically conducts all the activation patching experiments necessary to find which circuit composed of abstract units is responsible for the behavior. We have shown that ACDC and SP recover most of the compositional circuit that implements a language model behavior, as judged by comparison to previous mechanistic interpretability work (Section 4). ACDC with zero activations fully recovers the circuit of toy models (Fig. 9). Further, there is early evidence of the use of ACDC to help with novel interpretability work, discovering a surprising outline of a subgraph of GPT-2 Small that predicts gendered pronoun completion (Appendix K). Here, practitioners used ACDC to generate a subgraph including the most important pathway through a model's computation, and checked that this reflects the model's computation in normal (unablated) forward passes. This surprising find was an early example of the summarization motif (Tigges et al., 2023).

However, both ACDC and SP have limitations which prevent them from fully automating step 3 of the identified workflow (activation patching). First, they tend to miss some classes of abstract units that are part of the circuit, for example the negative name mover heads from IOI (Wang et al., 2023).

Second, the behavior of the algorithms is very sensitive to hyperparameter and metric choice, leading to varied and non-robust performance in some settings (Figure 3).

On balance, the evidence supports the claim that ACDC can automate part of interpretability work, a novel contribution. Automating interpretability research may be necessary to be able to scale methods to the behaviors of the large models which are in use today. We hope that our open-source implementation of ACDC (https://github.com/ArthurConmy/Automatic-Circuit-Discovery) accelerates interpretability research from the community. For example, future work could systematize and automate the problem of varying the corrupting dataset to understand the functionality of different parts of the circuit.

## 7 Acknowledgements

This work would not have been possible without the generous support of Redwood Research through their REMIX program. We would like to thank Chris Mathwin, Jett Janiak, Chris MacLeod, Neel Nanda, Alexandre Variengien, Joseph Miller, Thomas Kwa, Sydney von Arx, Stephen Casper and Adam Gleave for feedback on a draft of this paper. Arthur Conmy would like to thank Jacob Steinhardt, Alexandre Variengien and Buck Shlegeris for extremely helpful conversations that shaped ACDC. We would also like to thank Haoxing Du for working on an early tool, Nate Thomas for coming up with the catchy name, Daniel Ziegler who discussed experiments that inspired our Subnetwork Probing analysis, Oliver Hayman who worked on an earlier prototype during REMIX and Lawrence Chan who helped us frame our contributions and suggested several experiments. Finally we thank Hofvarpnir Studios, FAR AI and Conjecture for providing compute for this project.

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

# Appendix

# A  Table of contents

## B   Impact statement

ACDC was developed to automate the circuit discovery step of mechanistic interpretability studies. The primary social impact of this work, if successful, is that neural networks will become more interpretable. ACDC could make neural networks more interpretable via i) removing uninterpretable and insignificant components of models (as we reviewed in Section 4.2), ii) assisting practitioners to find subgraphs and form hypotheses for their semantic roles in models (as we found early evidence for in Appendix K) and more speculatively iii) enabling research that finds more interpretable architectures. More generally, better interpretability may allow us to predict emergent properties of models (Nanda et al., 2023), understand and control out-of-distribution behavior (Mu and Andreas, 2021) and identify and fix model errors (Hernandez et al., 2022).

However it is also possible that the biggest impact of better interpretability techniques will be more capable AI systems or possible misuse risk (Brundage et al., 2018). For example, while interpreting neural networks has the steer models towards model bias or other harmful effects (Cuadros, Zappella, and Apostoloff, 2022), bad actors could also use interpretability tools to do the opposite: reverse engineering neural networks to steer towards harmful behaviors.

For now, ACDC is a tool for researchers and isn't mature enough for applications where determining the exact behaviour of a model is societally important. However, the benefits of the adoption of better transparency appear to us to outweigh the externalities of potential negative outcomes (Hendrycks and Mazeika, 2022), as for example transparency plays an important role in both specific (Hubinger, 2020) and portfolio-based (Hendrycks, Mazeika, and Woodside, 2023) approaches to ensuring the safe development of AI systems.

# C  Discussion of metrics optimized

In this appendix, we discuss the considerations and experiments that support the formulation of ACDC that we presented in Section 3. We also discuss the metrics and experimental setups for Subnetwork Probing (Appendix D.1) and Head Importance Score for Pruning (Appendix D.2).

In the main text we presented ACDC as an algorithm that minimizes the KL divergence $D_{\mathrm{KL}}(G||H)$ between the model and the subgraphs of the model (Section 3 and Algorithm 1). However, prior mechanistic interpretability projects have reported performance on several different metrics at once (Wang et al., 2023; Nanda et al., 2023). In this Appendix we discuss our findings choosing different metrics in different ways. We explore the advantages and limitations with Algorithm 1 and other approaches. In particular, we have found that optimizing for low KL divergence is the simplest and most robust metric to optimize across different tasks. However, general conclusions about the best methods to use cannot be made because of variability across different tasks, and the large space of design choices practitioners can make.

We found that optimizing for low KL divergence was fairly effective across all tasks we considered, except the Docstring task (Appendix F-J). For example, we were able to exclusively recover heads that are present in the IOI circuit (Figure 1) that have 3 layers of composition sufficient to solve the task, with zero false positives . Additionally, KL divergence can be applied to any task of next-token prediction as it doesn't specify any labels associated with outputs (such as logit difference requiring specifying which tokens we calculate logit difference between).

## C.1  Changing the metric in ACDC

We consider generalizations of ACDC in order to further evaluate our patching-based circuit-finding approach. The only line of Algorithm 1 that we will modify is Line 6, the condition

$$D_{KL}(G||H_{\mathrm{new}}) - D_{KL}(G||H) < \tau \tag{1}$$

for the removal of an edge. All modifications to ACDC discussed in this Appendix replace Condition (1) with a new condition and do not change any other part of Algorithm 1.

In full generality we let $F$ be a metric that maps subgraphs to reals. We assume throughout this Appendix that subgraphs $H$, such that $F(H)$ is smaller, correspond to subgraphs that implement the task to a greater extent (i.e we minimize $F$).[5]

In practice, we can be more specific about the form that the metric $F$ will always take. We assume that we can always calculate $F(H(x_i, x_i'))$, the element-wise result of the metric on individual dataset examples and use $F(H)$ to refer to the metric averaged across the entire dataset (note the similarity of this setup to our calculation of $D_{\mathrm{KL}}$ in Section 3). The general update rule takes the form

$$F(H) - F(H_{\mathrm{new}}) < \tau. \tag{2}$$

which generalizes Equation 1. We discuss further extensions that change Line 6 of the ACDC algorithm in Appendix C.3.

## C.2  Limitations of logit difference

The IOI (Appendix F), Docstring (Appendix H) and Gendered Pronoun Identification work (Appendix K) originally used a variant of *logit difference* to measure the performance of subgraphs. Logit difference is the difference in logits for a correct output compared to a baseline incorrect output. Then, these works compare the change from the logit difference of the model to the logit difference of their circuit. However, unlike KL divergence, this metric is not always positive — logit difference for a circuit could be larger or smaller than the logit difference of the model, and so the change in logit difference could be positive or negative. We discuss issues that arise with this approach

---

[5]The logit difference and probability difference metrics used by the IOI, Greater-Than and Docstring tasks were intended to be maximised by the respective researchers (Table 1) so we consider negated versions of these metrics.

in Appendix C.3, the empirical performance decrease when using logit difference can be found in Figure 5.

## C.3 Alternatives to minimizing a metric

Two alternatives to minimizing a metric are to 1) match the model's performance on a metric, or 2) only include edges that cause a small change in performance. These could be formalised by the following alternatives to Condition 1, where $F$ denotes any metric we could compute from a subgraph:

1. **Matching the model's performance**: $|F(H_{\text{new}}) - F(G)| - |F(H) - F(G)| < \tau$.
2. **Only including small changes in performance**: $|F(H_{\text{new}}) - F(H)| < \tau$.

**Matching the model's performance** (also referred to as faithfulness by Wang et al. (2023)). Since KL divergence is always positive, Alternative 1 is identical to Condition 1 when $F$ is the KL divergence between a subgraph's outputs and the models' outputs, but for metrics such as logit difference this represents a new optimization objective. Empirically we found that matching the model's performance was unstable when we ran ACDC. For example, we ran a modified early version of ACDC that maximized the logit difference in the IOI circuit, and found that through one ACDC run, logit difference of a subgraph could be as large as 5.0 and as low as 1.5 during a modified ACDC run. The IOI circuit has a logit difference of 3.55, and therefore the subgraph's logit difference can be both larger and smaller than the model's logit difference. This issue arises when the subgraph's logit difference is larger than the model's. In such cases, ACDC will discard model components that it would otherwise include when the subgraph's logit difference is smaller than the model's. This leads to inconsistencies between runs and further dependence on the order over which parents are iterated (Section 3).

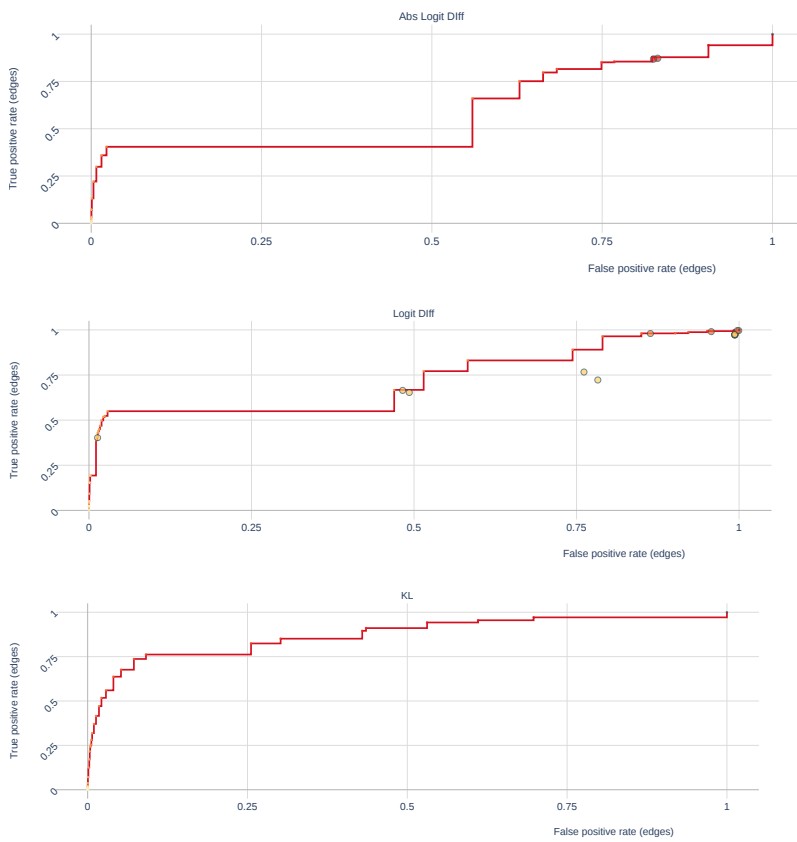

Figure 5: ROC curves on IOI using Abs Logit Diff, Logit Diff and KL Divergence

**Only including small changes in performance**. Alternative 2 ignores the value of the metric for the base model and instead focuses on how much the metric changes at each step that ACDC takes.

However, we found that this was much less effective than KL divergence and even worse than using logit difference on the IOI task (Figure 5), when we used random ablations and recorded ROC curves.

Overall, we found that KL divergence was the least flawed of all metrics we tried in this project, and think more practitioners should use it given i) the problems with other metrics listed, and ii) how empirically it can recover several circuits that were found by researchers using other metrics.

# D  Details of Subnetwork Probing and Head Importance Score for Pruning

## D.1  Subnetwork Probing

There are 3 modifications we made to Subnetwork Probing (Cao, Sanh, and Rush, 2021, SP) in our work. In this Appendix we provide techinal detail and motivation for these modifications:

1. **We do not train a probe**. ACDC does not use a probe. Cao, Sanh, and Rush (2021) train a linear probe after learning a mask for every component. The component mask can be optimized without the probe, so we just omit the linear probing step.
2. **We change the objective of the SP process to match ACDC's**. ACDC uses a task-specific metric, or the KL divergence to the model's outputs (Algorithm 1). In order to compare the techniques in equivalent settings we use the same metric (be it KL divergence or task-specific) in SP. Cao, Sanh, and Rush (2021) use negative log probability loss.
3. **We generalize the masking technique so we can replace activations with both zero activations and corrupted activations**. Replacing activations with zero activations[6] is useful for pruning (as they improve the efficiency of networks) but are not as commonly used in mechanisitic interpretability (Goldowsky-Dill et al., 2023), so we adapt SP to use corrupted activations. SP learns a mask $Z$ and then sets the weights $\phi$ of the neural network equal to $\phi * Z$ (elementwise multiplication), and locks the attention mask to be binary at the end of optimization (Jang, Gu, and Poole, 2017). This means that outputs from attention heads and MLPs in models are scaled closer to 0 as mask weights are decreased. To allow comparison with ACDC, we can linearly interpolate between a clean activation when the mask weight is 1 and a corrupted activation (i.e a component's output on the datapoint $x_i'$, in the notation of Section 3) when the mask weight is 0. We do this by editing activations rather than weights of the model.

Additionally, we used a constant learning rate rather than the learning rate scheduling used in Cao, Sanh, and Rush (2021).

The regularization coefficients $\lambda$ (in the notation of Cao, Sanh, and Rush (2021)) we used in Figure 4 were 0.01, 0.0158, 0.0251, 0.0398, 0.0631, 0.1, 0.158, 0.251, 0.398, 0.631, 1, 2, 3, 4, 5, 6, 7, 8, 9, 10, 30, 50, 70, 90, 110, 130, 150, 170, 190, 210, 230, 250.

The number of edges for subgraphs found with Subnetwork Probing are computed by counting the number of edges between pairs of unmasked nodes.

## D.2  Head Importance Score for Pruning

In Section 4-4.2 we compared ACDC with the Head Importance Score for Pruning (Michel, Levy, and Neubig, 2019, HISP). We borrow the author's notation throughout this section, (particularly from Section 4.1) and adapt it so that this Appendix can be read after Section 2.

The authors use masking parameters $\xi_h$ for all heads, i.e scaling the output of each attention head by $\xi_h$, similar to the approach in Subnetwork Probing (Appendix D.1), so that each head $\text{Att}_h$'s output is $\xi_h \text{Att}_h(x_i)$ on an input $x_i$. The authors keep MLPs present in all networks that they prune. We generalize their method so that it can be used for any component inside a network for which we can take gradients with respect to its output.

The authors define head importance as

$$I_h := \frac{1}{n} \sum_{i=1}^{n} \left| \frac{\partial \mathcal{L}(x_i)}{\partial \xi_h} \right| = \frac{1}{n} \sum_{i=1}^{n} \left| \text{Att}_h(x_i)^T \frac{\partial \mathcal{L}(x_i)}{\partial \text{Att}_h(x_i)} \right|. \tag{3}$$

---

[6]Which is generally equivalent to setting weight parameters equal to 0.

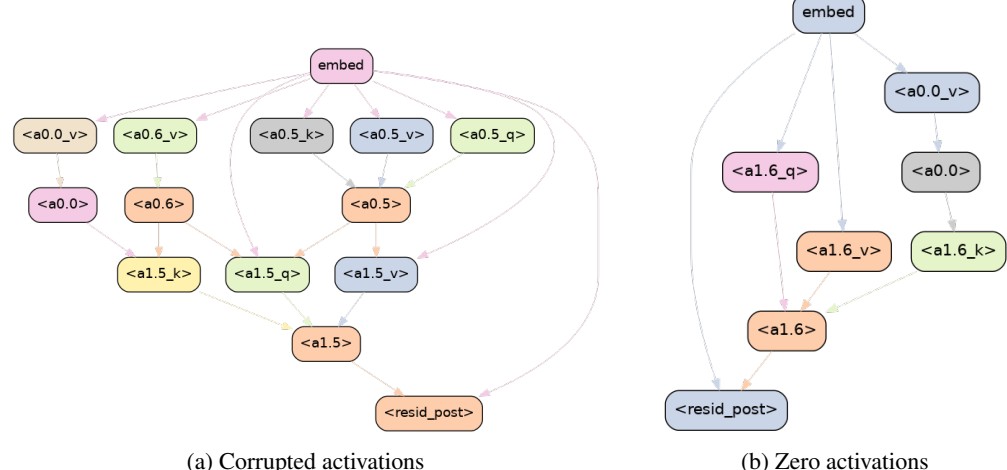

(a) Corrupted activations          (b) Zero activations

Figure 6: Examples of subgraphs recovered by ACDC on the induction task with different types of activations and threshold $\tau = 0.5623$. These find the two different induction heads (1.5, 1.6) and a previous token head (0.0) as identified by Goldowsky-Dill et al. (2023). (a) shows the result with corrupted activations, while (b) shows the result with zero activations.

where the equivalence of expressions is the result of the chain rule. We make three changes to this setup to allow more direct comparison to ACDC: i) we use a metric rather than loss, ii) we consider corrupted activations rather than just zero activations and iii) we use the 'head importance' metric for more internal components than merely attention head outputs.

Since our work uses in general uses a metric $F$ rather than loss Appendix C, we instead use the derivative of $F$ rather than the derivative of the loss. The HISP authors only consider interpolation between clean and zero activations, so in order to compare with corrupted activations, we can generalize $\xi_h$ to be the interpolation factor between the clean head output $\text{Att}_h(x)$ (when $\xi_h = 1$) and the corrupted head output $\text{Att}_h(x')$ (when $\xi_h = 0$). Finally, this same approach works for any internal differentiable component of the neural network.[7] Therefore we study the HISP applied to the query, key and value vectors of the model and the MLPs outputs.

In practice, this means that we compute component importance scores

$$I_C := \frac{1}{n} \sum_{i=1}^{n} \left| (C(x_i) - C(x_i'))^T \frac{\partial F(x_i)}{\partial C(x_i)} \right|. \tag{4}$$

Where $C(x_i)$ is the output of an internal component $C$ of the transformer, which is equivalent to 'attribution patching' (Nanda, 2023) up to the absolute value sign.

To compute the importance for zero activations, we adjust Equation (4) so it just has a $C(x_i)$ term, without the $-C_h(x_i')$ term. We also normalize all scores for different layers as in Michel, Levy, and Neubig (2019). The identical setup to Equation (4) works for outputs of the query, key and value calculations for a given head, as well as the MLP output of a layer. In Section 4 we use query, key and value components for each head within the network, as well as the output of all MLPs.

The number of edges for subgraphs found with HISP is also computed by counting the number of edges between pairs of unmasked nodes, like Subnetwork Probing (Appendix D.1).

# E   Experimental study of algorithm design

This section evaluates design choices for ACDC and SP, by re-doing the experiments in Section 4. We explore two axes of variation.

---

[7]In theory. In practice, components need be `torch.nn.Modules` such that we can calculate the gradient of $F$ with respect to the components' outputs.

Table 2: AUCs for corrupted activations, Random Ablation. (E)=Edge, (N)=Node.

| Metric | Task | ACDC(E) | HISP(E) | SP(E) | ACDC(N) | HISP(N) | SP(N) |
|---|---|---|---|---|---|---|---|
| KL | Docstring | **0.982** | 0.805 | 0.937 | **0.950** | 0.881 | 0.928 |
| | Greaterthan | **0.853** | 0.693 | 0.806 | **0.890** | 0.642 | 0.827 |
| | IOI | **0.869** | 0.789 | 0.823 | **0.880** | 0.668 | 0.842 |
| Loss | Docstring | **0.972** | 0.821 | 0.942 | 0.938 | 0.889 | **0.941** |
| | Greaterthan | 0.461 | 0.706 | **0.812** | 0.766 | 0.631 | **0.811** |
| | IOI | 0.589 | **0.836** | 0.707 | 0.777 | 0.728 | **0.797** |
| | Tracr-Proportion | **0.679** | **0.679** | 0.525 | 0.750 | **0.909** | 0.818 |
| | Tracr-Reverse | 0.200 | **0.577** | 0.193 | 0.312 | **0.750** | 0.375 |

Table 3: AUCs for corrupted activations, Zero Ablation. (E)=Edge, (N)=Node.

| Metric | Task | ACDC(E) | HISP(E) | SP(E) | ACDC(N) | HISP(N) | SP(N) |
|---|---|---|---|---|---|---|---|
| KL | Docstring | **0.906** | 0.805 | 0.428 | 0.837 | **0.881** | 0.420 |
| | Greaterthan | **0.701** | 0.693 | 0.163 | **0.887** | 0.642 | 0.134 |
| | IOI | 0.539 | **0.792** | 0.486 | 0.458 | **0.671** | 0.605 |
| Loss | Docstring | **0.929** | 0.821 | 0.482 | 0.825 | **0.889** | 0.398 |
| | Greaterthan | 0.491 | **0.706** | 0.639 | **0.783** | 0.631 | 0.522 |
| | IOI | 0.447 | **0.836** | 0.393 | 0.424 | **0.728** | 0.479 |
| | Tracr-Proportion | **1.000** | 0.679 | 0.829 | **1.000** | 0.909 | **1.000** |
| | Tracr-Reverse | **1.000** | 0.577 | 0.801 | **1.000** | 0.750 | **1.000** |

- Minimizing the task-specific metric, rather than the KL divergence.
- Patching activations with zeros, rather than with the result on a corrupted input (interchange intervention).
- Looking at node-level TPR and FPR for the ROC curves, rather than edge-level.

The results paint a mixed picture of whether ACDC or SP is better overall, but reinforce the choices we implicitly made in the main text. A stand-out result is that ACDC with zero-patching is able to perfectly detect the `tracr` circuits (Figs. 9 and 10).

A numerical summary of the results is in Tables 2 and 3, which display the areas under the ROC curve (AUC) for all the design choices we consider.

### E.1 Minimizing the task-specific metric, rather than the KL divergence

We ran ACDC, SP and HISP with the task-specific metric from Table 1, instead of KL divergence. The exact modification is described in Appendix C. The ROC result is in Fig. 7. Compared to minimizing KL divergence (Fig. 3), ACDC works better for Docstring, but worse for Greater-Than and IOI, indicating that it is not a very robust method.

We prefer using the KL divergence instead of the task-specific metric, because the task-specific metric can be over-optimized (Appendix C). This means that the recovered circuit ends up performing the task more than the original model, and is thus not accurate. We can observe this effect by comparing the task-specific metric achieved by the methods in Fig. 8.

### E.2 Activation patching with zeros, instead of corrupted input

In the main text experiments that compared using corrupted activations and zero activations (Figure 4), all three methods recovered subgraphs with generally lower loss when doing activation patching with zeros, in both the experiments with the normal model and with permuted weights. It is unclear why the methods achieve better results with corruptions that are likely to be more destructive. A possible explanation is that there are 'negative' components in models (Appendix F.3) that are detrimental to the tasks, and the zero activations are more disruptive to these components. A discussion of how methods could be adjusted to deal with this difficulty can be found in Alternative 2 in Appendix C.

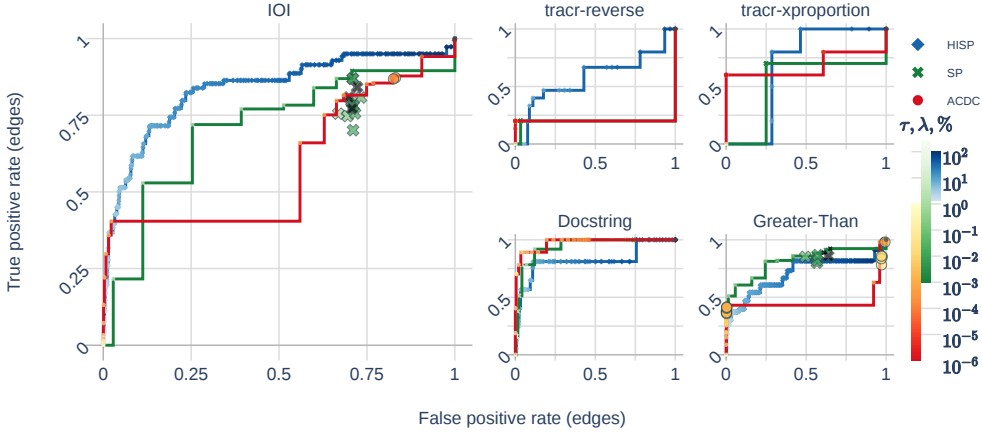

Figure 7: Edge-wise ROC curves generated by minimizing the task-specific metric in Table 1, rather than KL divergence.

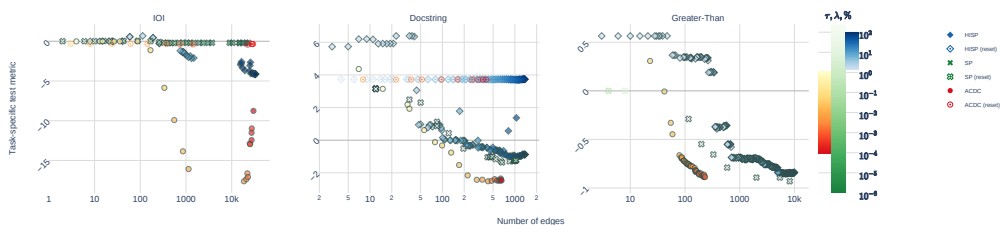

Figure 8: Optimizing the task-specific metric of the subject model, on trained and reset networks. For each recovered circuit, we plot its task-specific metric (Table 1) against its number of edges. The reset networks metrics don't change much with the number of edges, which is good. We found that for IOI that extremely large logit differences could be achieved (over 15) but this didn't happen when the network had a large number of edges.

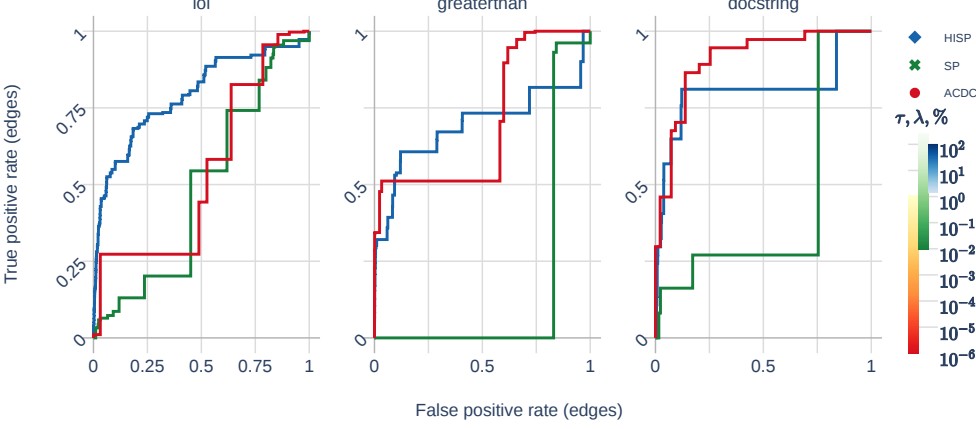

Figure 9: Edge-wise ROC curves generated by minimizing the KL divergence, but using zero activations.

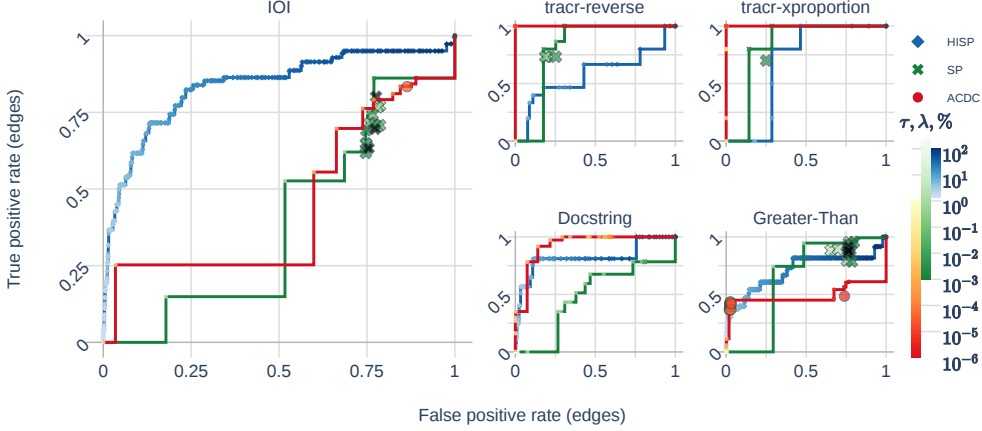

Figure 10: Edge-wise ROC curves generated by minimizing the task-specific metric in Table 1, using zero activations.

### E.3 Node-level ROC curve, rather than edge-level ROC curve

We compute the FPR and TPR of classifying whether a node belongs to the circuit. We consider this alternative task because SP and HISP operate at the node-level, whereas ACDC operates at the edge-level, so this is fairer to HISP and SP. The results are broadly similar to edge-level ROCs, and are described in Figs. 11 and 12 and Tables 2 and 3.

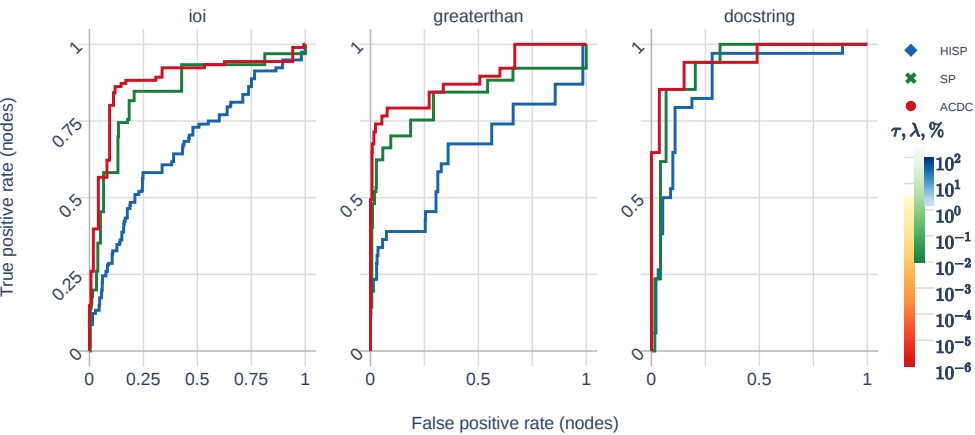

Figure 11: Node-level ROC when optimizing the KL divergence, with corrupted activations.

## F    IOI task: details and qualitative evidence

### F.1    Further details on the IOI experiments

In the ACDC run in Figure 1, we used a threshold of $\tau = 0.0575$. We also removed all edges which did not lie on a directed path from the input (which is equivalent in computation since we use corrupted activations). Our library now only supports splitting query, key and input, rather than merely looking at the connections between heads. Additionally, For ease of visualization, in the diagram on the left of Figure 1 we removed all edges between grey nodes more than 2 layers apart, and 90% of the edges between grey and red nodes.

Our IOI experiments were conducted with a dataset of $N = 50$ text examples from one template the IOI paper used (' When John and Mary went to the store, Mary gave a bottle of milk to'). The

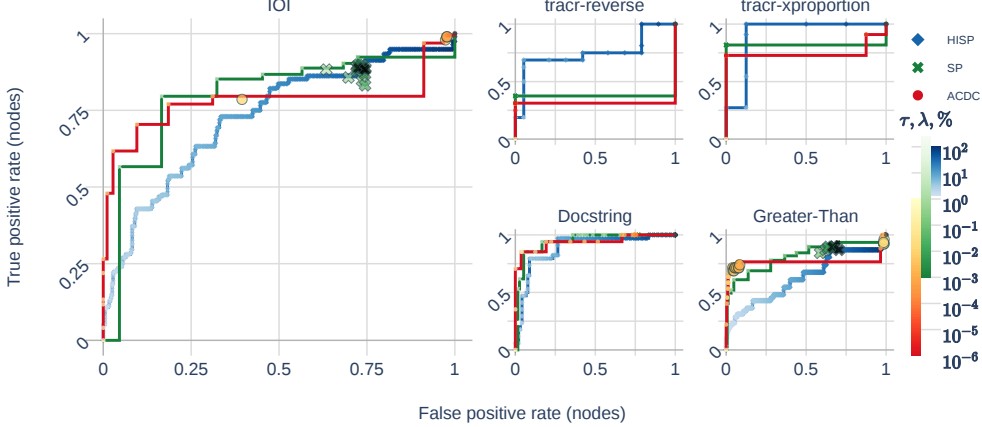

Figure 12: Node-level ROC when optimizing the task-specific metric in Table 1, with corrupted activations.

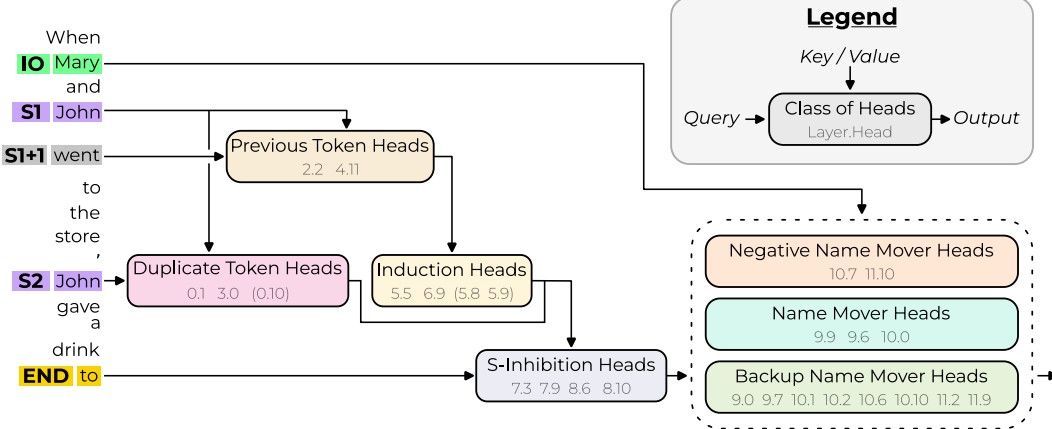

Figure 13: The IOI circuit (Figure 2 from Wang et al. (2023)). We include edges between all pairs of heads connected here, through the Q or the K+V as indicated. We also include all connections between MLPs and these heads, the inputs and outputs. See https://github.com/ArthurConmy/Automatic-Circuit-Discovery/blob/main/acdc/ioi/utils.py#L205. The full circuit is in Fig. 14.

corrupted dataset was examples from the ABC dataset (Wang et al., 2023) — for example 'When Alice and Bob went to the store, Charlie gave a bottle of milk to'.

In the IOI experiment in Figure 1, we did not split the computational graph into the query, key and value calculations for each head. This enabled the ACDC run to complete in 8 minutes on an NVIDIA A100 GPU. However, the larger experiments that kept >10% of the edges of the original edges in the computational graph sometimes took several hours. On one hand we don't expect these cases to be very important for circuit discovery, but they make up the majority of the points of the pareto frontier of curves in this paper.

### F.2 The IOI circuit

Wang et al. (2023) find a circuit 'in the wild' in GPT-2 small (Radford et al., 2019). The circuit identifies indirect objects (see for example Table 1) by using several classes of attention heads. In this subsection we analyze how successful ACDC's circuit recovery (Figure 1) is. All nine heads found in Figure 1 belong to the IOI circuit, which is a subset of 26 heads out of a total of 144 heads in GPT-2 small. Additionally, these 9 heads include heads from three different classes (Previous Token Heads,

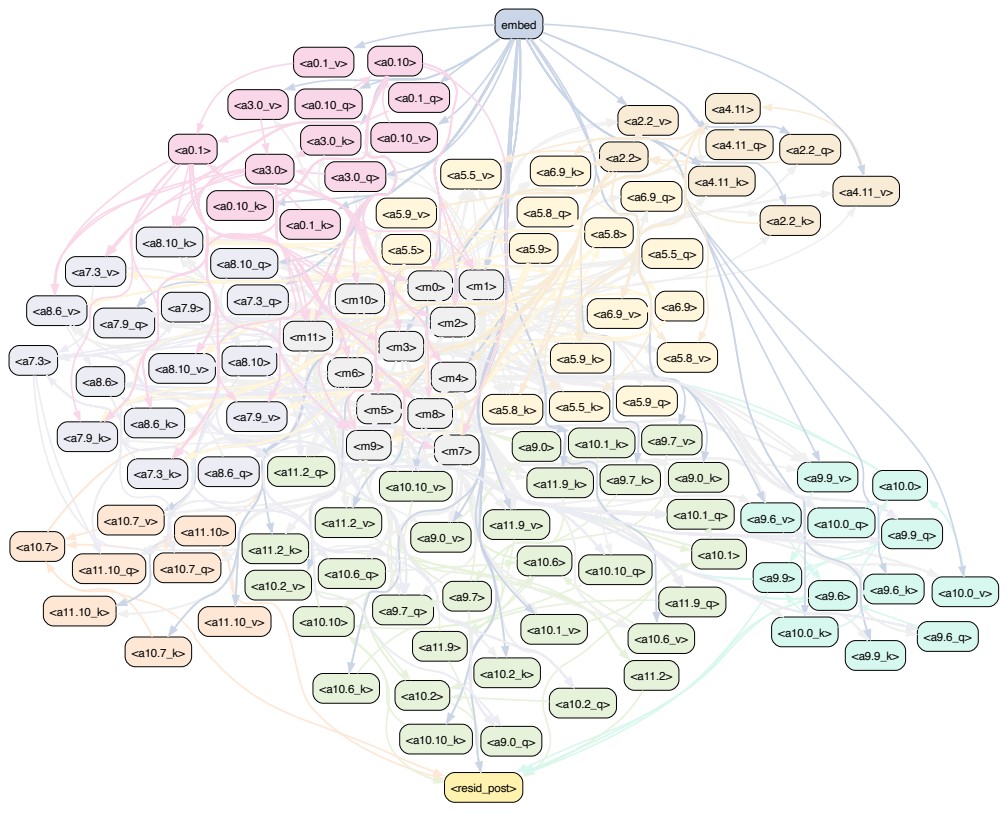

Figure 14: Our low-level implementation of the IOI circuit (Wang et al., 2023), in terms of heads split by query, key, value; and MLPs. It has 1041 edges. For edge between groups $A$, $B$ in Fig. 13, we connect each member of group $A$ with each member of group $B$. The colors of groups in this figure correspond to the group colors in Fig. 13.

S-Inhibition Heads and Name Mover Heads) and are sufficient to complete the IOI task, showing that ACDC indeed can recover circuits rather than just subgraphs.

For our ROC plots, we considered the computational graph of IOI described in Figure 13.

The ground-truth circuit gets a logit difference of 3.24 compared to the model's 4.11 logit difference. It has a KL divergence of 0.44 from the original model.

## F.3 Limitations of ACDC in recovering the IOI circuit

The main figure from the paper Figure 1 shares several features with circuits recovered with similar thresholds, even when logit difference rather than KL divergence is minimized. The figure does not include heads from all the head classes that Wang et al. (2023) found, as it does not include the Negative Name Mover Heads or the Previous Token Heads. In Figure 15 we run ACDC with a lower threshold and find that it does recover Previous Token Heads and Negative Name Mover Heads, but also many other heads not documented in the IOI paper. This is a case where KL divergence performs better than logit difference maximisation (which does not find Negative Name Movers at any threshold), but still is far from optimal (many extraneous heads are found). Ideally automated circuit discovery algorithms would find negative components even at higher thresholds, and hence we invite future empirical and theoretical work to understand negative components and build interpretability algorithms capable of finding negative components. An early case study gaining wide understanding of a negative head in GPT-2 Small can be found in McDougall et al. (2023).

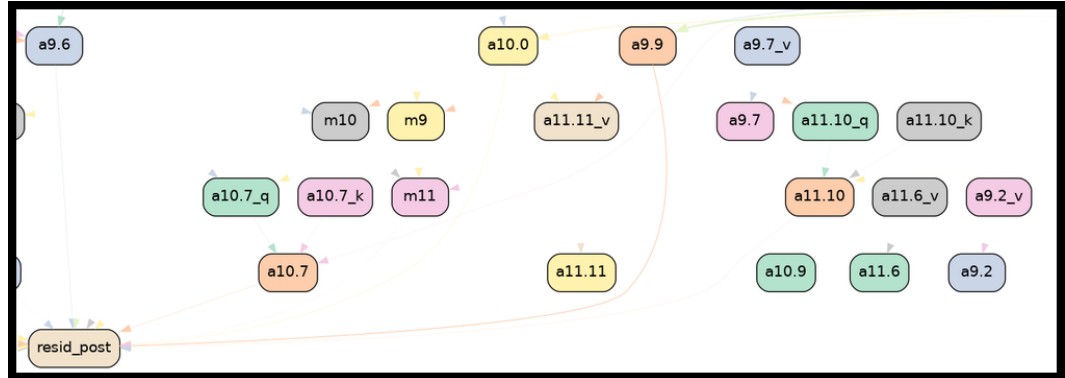

Figure 15: A subset of the 443/32923 edges of GPT-2 Small that ACDC recovered when optimizing for KL divergence at threshold $\tau = 0.00398$. This subset includes edges between Negative Heads (10.7 and 11.10). A number of heads not found by the IOI work (9.2 and 11.11 for example) were also found.

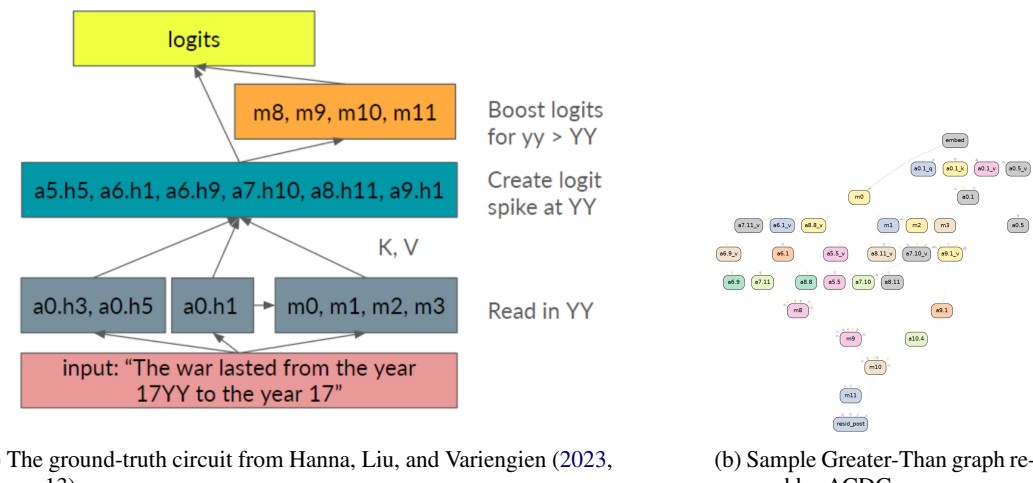

(a) The ground-truth circuit from Hanna, Liu, and Variengien (2023, Figure 13).

(b) Sample Greater-Than graph recovered by ACDC

Figure 16: The ground-truth Greater-Than circuit (16a) and a circuit that ACDC recovers (16b).

# G  Greater-Than task: details and qualitative evidence

We use a random sample of 100 datapoints from the dataset provided by Hanna, Liu, and Variengien (2023).

We use the circuit from Figure 13 from their paper, including connections between MLPs that are in the same group (e.g MLPs 8, 9, 10 and 11) but not including connections connections between attention heads in the same group. We also include all Q and K and V connections between attention heads present. Their circuit includes all earlier layer connections to the queries of the mid-layer attention heads that cause a logit spike (Figure 16a). This would account for more than half of the edges in the circuit were we to include all such edges that compute these query vectors, and hence we compromised by just adding all early layer MLPs as connections to these query vectors. Full details on the circuit can be found in our codebase[8] and in Fig. 17. This circuit gets a probability difference score of 72% on a subset of their dataset for which GPT-2 Small had an 84% probability difference and a KL divergence from the original model of 0.078.

[8]https://github.com/ArthurConmy/Automatic-Circuit-Discovery/blob/main/acdc/greaterthan/utils.py#L231

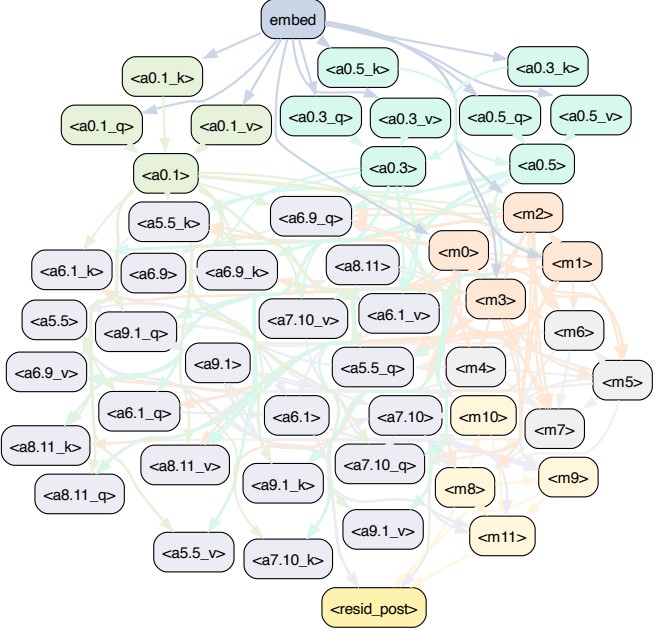

Figure 17: Our low-level implementation of the Greater-Than circuit (Hanna, Liu, and Variengien, 2023), in terms of heads split by query, key, value; and MLPs. It has 262 edges. For edge between groups $A$, $B$ in Fig. 16a, we connect each member of group $A$ with each member of group $B$.

An example subgraph ACDC recovered, including a path through MLP 0, mid-layer attention heads and late-layer MLPs, is shown in Figure 16b. This run used the Greater-Than probability difference metric and a threshold of 0.01585 and recovered the results in the abstract.

## H   Docstring task: details and qualitative evidence

### H.1   The docstring circuit

Heimersheim and Janiak (2023) find a circuit in a small language model that is responsible for completing Python docstrings. The model is a 4-layer attention-only transformer model, trained on natural language and Python code. This circuit controls which variable name the model predicts in each docstring line, e.g for the prompt in Table 1 it chooses shape over the other variable names files, obj, state, size, or option.

The circuit is specified on the level of attention heads, consisting of 8 main heads (0.2, 0.4, 0.5, 1.4, 2.0, 3.0, and 3.6) that compose over four layers, although it only makes use of three levels of composition. It consists of 37 edges between inputs, output, and the attention heads, as we show in Figure 18a. We discuss below why we exclude 0.2 and 0.4.

We apply the ACDC algorithm (Section 2) to this dataset, using the prompts from Heimersheim and Janiak (2023) found in their accompanying Colab notebook.[9] For our corrupted dataset we use their random_random dataset which randomizes both the variable names in the function definition as well as in the docstring of prompts.

ACDC generates the subgraph shown in Figure 18b. We now compare this to the original 8-head circuit from Heimersheim and Janiak (2023), which was the most specific circuit *evaluated* in that work. We refer to this circuit as the 'manual' circuit to distinguish it from the ground truth, which

---

[9]Available at https://colab.research.google.com/drive/17CoA1yARaWHvV14zQGcI3ISz1bIRZKS5 as of 8th April 2023

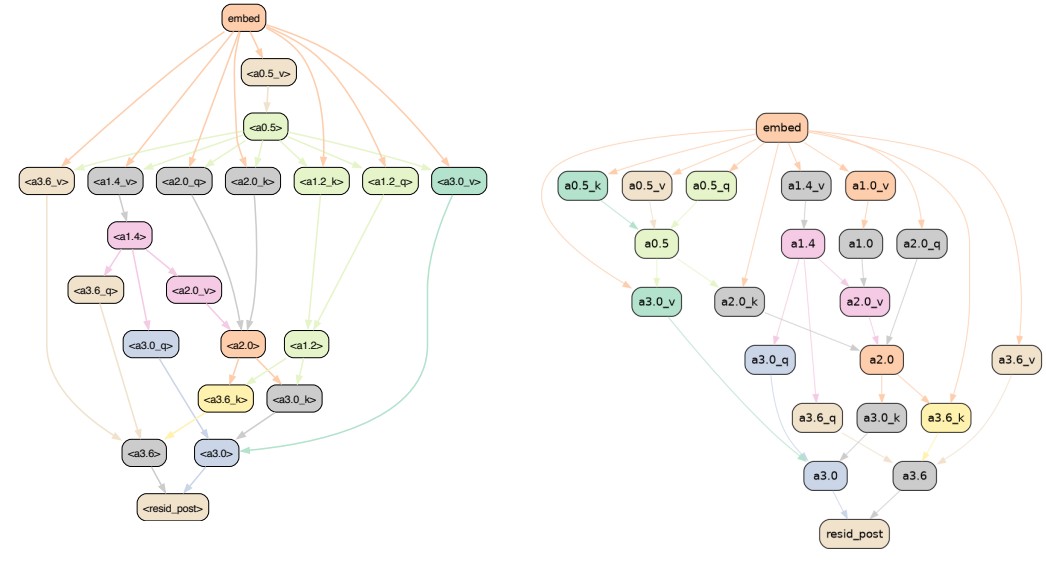

(a) Canonical Docstring circuit (37 edges)     (b) ACDC circuit (KL-divergence, $\tau = 0.095$)

Figure 18: Our implementation of the Docstring circuit (Heimersheim and Janiak, 2023), compared to an ACDC-generated circuit.

| Metric | Full model | ACDC KL $\tau = 0.005$ | ACDC KL $\tau = 0.095$ (Fig. 18b) | ACDC LD $\tau = 0.067$ (Fig. 19) | Manual 8 heads, all connections | Ground-truth circuit (Fig. 18a) |
|---|---|---|---|---|---|---|
| KL-divergence | 0 | 0.33 | 1.2 | 0.67 | 0.83 | 1.1 |
| Mean logit diff. | 0.48 | 0.58 | -1.7 | 0.32 | -0.62 | -1.6 |
| Num. of edges | 1377 | 258 | 34 | 98 | 464 | 37 |

Table 4: Comparing our ACDC docstring results to the ground-truth from Heimersheim and Janiak (2023) using their metrics. We compare (from left to right) the full model, the subgraph from ACDC runs optimizing for KL divergence ($\tau = 0.005$ and $0.095$) and logit difference ($\tau = 0.067$), as well as the two subgraphs made manually from Heimersheim and Janiak (2023): One including all connections between the given attention heads, and one using only the given circuit. The metrics used are KL divergence between full-circuit outputs and resample-ablated output (lower is better), mean logit difference between correct and wrong completions (higher is better), and the number of edges in the circuit (lower is better).

includes the edge connections that the authors speculated were most important but did not evaluate due to a lack of software for edge-editing. We find (a) overlapping heads, (b) heads found by ACDC only, and (c) heads found in the manual interpretation only. In the first class (a) we find heads 0.5, 1.4, 2.0, 3.0, and 3.6. All these manually identified heads are recovered by ACDC. In class (b) we find head 1.0 which the authors later add to their circuit to improve performance; ACDC shows for the first time where this head belongs in the circuit. In class (c) we find heads 0.2, 0.4 and 1.2. However, the first two of these are not actually relevant under the docstring distribution and only added by the authors manually. Head 1.2 is considered a non-essential but supporting head by the authors and not identified by ACDC at the chosen threshold of $\tau = 0.095$ (for KL divergence). This might be because 1.2 is less important than the other heads, and indeed we recover this head in larger subgraphs (such as the subgraph in Figure 19).

We compare the numerical results between the ACDC circuits and the circuit described in Heimersheim and Janiak (2023) in Table 4. In addition to the $\tau = 0.095$ run (Figure 18b) we perform a run with lower KL divergence threshold of $\tau = 0.005$ recovering a larger circuit (258 edges) containing also head 1.2 that was missing earlier.

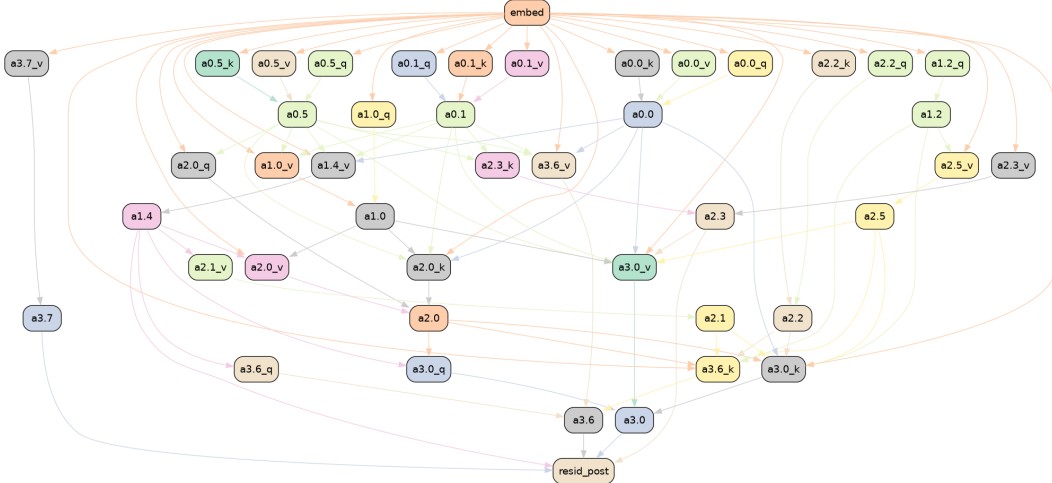

Figure 19: ACDC-found subgraph for docstring task minimizing logit difference ($\tau = 0.067$) instead of KL divergence.

Since Heimersheim and Janiak (2023) use logit difference as their metric, we add an ACDC run that optimizes logit difference rather than KL divergence (see Appendix C for details on this adjustment) with threshold $\tau = 0.067$. This circuit (Figure 19) recovers the relevant manual-interpretation heads (including 1.2) as well.[10] It is even more specific, containing 93% less edges than the full circuit. This is also 79% less edges than the head-based circuit from Heimersheim and Janiak (2023) while achieving a better score on all metrics.

Note that there are two versions of the manual circuit we consider. There is (i) the set of 8 heads given in Heimersheim and Janiak (2023) that the authors test with a simple methods (not specifying edges), and (ii) the circuit of 39 edges as suggested by the authors that they were not able to test due to not having software to implement editable transformer computational graphs in PyTorch. We reconstruct this circuit, shown in Figure 18a, from their descriptions and perform tests (Table 4).

In case (i) the ACDC run (threshold $\tau = 0.005$) achieves better performance in both metrics, Logit Difference and KL divergence, while being more specific (258 edges) when compared to the set of heads found by Heimersheim and Janiak (2023). In the more specific case (ii) the ACDC run (with threshold $\tau = 0.095$) closely matches the manual interpretation, with a very similar circuit recovered (Figure 18). The ACDC run is slightly more specific but has slightly worse KL divergence and Logit Difference.

A limitation worth noting is that we applied ACDC to a computational graph of attention heads and their query, key and value computational nodes, while Heimersheim and Janiak (2023) considered the attention heads outputs into every token position separately. This allowed them to distinguish two functions fulfilled by the same attention head (layer 1, head 4) at different positions, which cannot be inferred from the output of ACDC alone at any level of abstraction (Section 2.2) we studied in this work. We make this choice for performance reasons (the long sequence length would have made the experiments significantly slower) but this is not a fundamental limitation. In Appendix K we use ACDC to isolate the effects of individual positions in a different task.

### H.2 Additional docstring experiments

**Logit difference metric:** To compare ACDC more closely with the docstring work (Heimersheim and Janiak, 2023), we added an ACDC run with the objective to maximize the logit difference metric. We used a threshold of $\tau = 0.067$ and found the subgraph shown in Figure 19. We found that ACDC performed better than SP and HISP when using the logit difference metric (Figure 7).

**Zero activations:** Unlike in the case of induction (Section 4.2), we found that using zero activations rather than random (corrupted) activations, lead to far worse results. For example, with $\tau = 0.067$

---

[10]Again, not considering heads 0.2 and 0.4 which are not actually relevant under the docstring distribution.

(the same threshold that generated Figure 19 except with zero activations) we get a circuit with 177 edges (Figure 19 has 98), as well as a KL divergence of 3.35 and a logit difference of $-2.895$. All these metrics are worse than the subgraphs generated with corrupted activations (Table 4).

## I  Tracr tasks: details and qualitative evidence

In this Appendix we discuss the two tracr tasks we studied in Section 4, as well as additional experiments that studied ACDC when applied at a neuron level.

We used a transformer identical to the one studied in Lindner et al. (2023), and refer to that work for details on the tracr-xproportion task (called the `frac_prevs` task in their paper). We also studied the tracr-reverse task, described in the tracr Github repository.[11]

We make one modification to the traditional ACDC setup. We set the positional embeddings equal to randomized positional embeddings in the corrupted datapoints — otherwise, we don't recover any of the circuit components that depend only on positional embeddings (and not token embeddings). We describe the two tasks that we studied in the main text and describe futher results that broke these computational graphs down into neurons.

### I.1  tracr-xproportion

We used the proportion task from the tracr main text, and used as metric the L2 distance between the correct list of proportions and the recovered list of proportions. For the corrupted dataset, we let $(x_i')_{i=1}^n$ be a random permuation of $(x_i)_{i=1}^n$ with no fixed points.

When we ran ACDC at the neuron level, as shown in Figure 20b, there are no extra nodes present that were not used by this tracr model. In fact, this computational graph visualization produced by ACDC is more compact than the complete view of the states of the residual stream which is illustrated in Figure 20a (from Lindner et al. (2023)). In this case, the transformer is small enough for a practitioner to study each individual state in the forward pass. However, for larger transformers this would be intractable, which necessitates the use of different interpretability tools such as ACDC.

See Fig. 21a for the full circuit (without decomposition into residual stream dimensions).

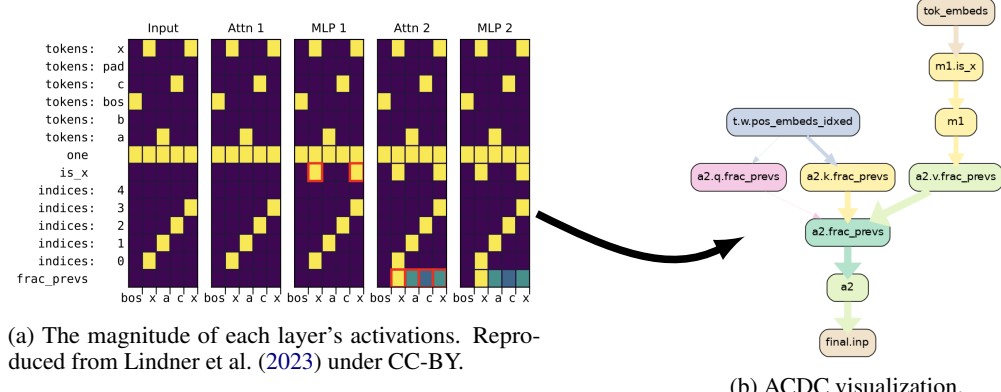

(a) The magnitude of each layer's activations. Reproduced from Lindner et al. (2023) under CC-BY.

(b) ACDC visualization.

Figure 20: Two visualizations of how a `tracr`-compiled transformer completes the `frac_prevs` task. The ACDC circuit is specific to the individual neurons and residual stream components. This is more fine-grained than the ground truth we use throughout the work. This experiment was coded in `rust_circuit` and is not reproducible using the Transformer Lens code yet.

### I.2  tracr-reverse

To test ACDC on more than one example of a tracr program, we also used the 3-layer transformer that can reverse lists (the tracr-reverse task). Once more, the outputs of the transformer are not

---

[11]URL: https://github.com/deepmind/tracr, file: README.md

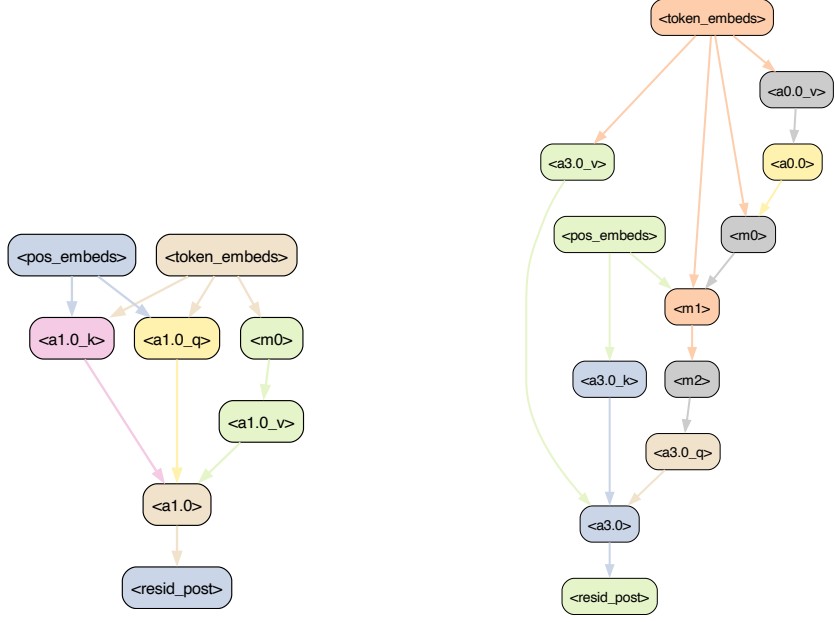

(a) tracr-xproportion canonical circuit (10 edges)          (b) tracr-reverse canonical circuit (15 edges)

Figure 21: The canonical circuits for finding the proportion of 'x' in the input and reversing lists. ACDC recovers these perfectly using zero activations (Table 3 and Figs. 9 and 10).

distributions - in this case they are new lists. We calculate the L2 distance between the one-hot vectors for the recovered list and the true reversed list. For the corrupted dataset, we again let $(x_i')_{i=1}^n$ be a random permuation of $(x_i)_{i=1}^n$ with no fixed points. Again, at the neuron level a perfect graph is recovered, with the minimal components required to reverse lists (Figure 21b).

## J  Induction task: details and qualitative evidence

In Section 4.2 we use 40 sequences of 300 tokens from a filtered validation set of OpenWebText (Gokaslan et al., 2019). We filter the validation examples so that they all contain examples of induction — subsequences of the form "$A, B, \ldots, A, B$", where $A$ and $B$ are distinct tokens. We only measure KL divergence for the model's predictions of the second $B$ tokens in all examples of the subsequences $A, B, \ldots, A, B$.

We use both zero activations and corrupted activations to compare ACDC and the other methods. To use ACDC with zero activations, we apply one change to the procedure described in Section 3: instead of setting activations of edges not present in the subgraph to the activations on a corrupted dataset, we set their value equal to 0. We describe how we adapt the methods from Section 4.1 to be used with both zero activations and corrupted activations in Appendix D.1 for SP and Appendix D.2 for HISP.

Our induction experiments were performed on a 2-layer, 8-head-per-layer attention only transformer trained on OpenWebText (Gokaslan et al., 2019). The model is available in the TransformerLens (Nanda, 2022) library.[12] We follow Appendix C of Goldowsky-Dill et al. (2023) for the construction of the dataset of induction examples.

The computational graph has a total of 305 edges, and in Figure 4 we only show subgraphs with at most 120 edges.

---

[12]The model can be loaded with `transformer_lens.HookedTransformer.from_pretrained(model_name = "redwood_attn_2l", center_writing_weights = False, center_unembed = False)` (at least for the repository version of the source code as of 23rd May 2023)

When iterating over the parents of a given node (Line 4 in Algorithm 1), we found that iterating in increasing order of the head index was important to achieve better results in Figure 4. Similar to all experiments in the work, we iterate in decreasing order of layers, so overall we iterate over head 1.0, 1.1, ... then 1.7, then 0.0, 0.1, ... .

An example of a circuit found in the process is given in Figure 6.

## K  Gendered pronoun completion: qualitative evidence

Mathwin et al. (2023) aim to isolate the subgraph of GPT-2 small responsible for correctly gendered pronouns in GPT-2 small. They do this by studying prompts such as "So Dave is a really great friend, isn't" which are predicted to finish with " he". For that they used ACDC. This presents an example of a novel research project based on ACDC. The result of applying the ACDC algorithm (threshold $\tau = 0.05$) is shown in Figure 22.

The computational subgraphs generated by ACDC on the gendered pronoun completion task show that MLP computations are more important than attention head computations in this task than in the IOI task (Appendix F.2). Early, middle and late layer MLPs have important roles in the subgraph. For example, MLPs 3 and 5 are the important components at the name position (which must be used to identify the correct gender) as they have multiple incident edges: the MLP 7 at the "_is" position has the most incoming connections of any node in the graph, and the late layer MLPs 10 and 11 have the largest direct effect on the output. MLP 7's importance at the "_is" position is an example of a discovery that could not have been made with simpler interpretability tools such as saliency maps. This was early evidence of the summarization motif (Tigges et al., 2023).

ACDC's output shows that the important internal information flow for predicting the correct gender has three steps. Firstly, Layer 0 attention head 0.4 and MLP0 use the name embedding which they pass (through intermediary MLPs) via key- and value-composition (Elhage et al., 2021) to attention heads 4.3 and 6.0. Secondly, heads 4.3 and 6.0 attend to the name position to compose with 0.4 and MLP0. Finally, through value-composition with attention heads 6.0 and 4.3 (via MLP7), the outputs of 10.9 and 9.7 output the expected gendered completion to the output node. Mathwin et al. (2023) then verified that indeed in a normal forward pass of GPT-2 Small, 0.4 has an attention pattern to itself at the name token, attention heads 4.3 and 6.0 attend to the previous name token, and 10.9 and 9.7 attend to the ' is' token. They also perform path patching experiments on intermediate nodes to provide further evidence of the importance of the pathway through the ' is' token.

We used the dataset of $N = 100$ examples from Mathwin et al. (2023) . The corrupted dataset was a set of prompts with a similar structure to the sentence "That person is a really great friend, isn't", following the authors' approach.

We defined a computational graph that featured nodes at the specificity of attention heads split by query, key and value vectors, and further split by token position (where the tokens are present in the nodes in Figure 22). From the input sentence 'So Sarah is a really nice person, isn't', we chose to add nodes representing the model internal operations at the tokens "_Sarah", "_is", "_person", "_isn" and "'t", while other positions were grouped together as in Mathwin et al. (2023) . The resulting subgraph can be found in Figure 22.

## L  Reset Network Experiments

Our **reset network** experiment setup is motivated by the concern that interpretability explanations may not accurately represent the reasoning process behind models' predictions (Jacovi and Goldberg, 2020). This is particularly relevant to work on subnetworks as empirically some subnetworks in models with randomized weights do not accurately represent such reasoning (Ramanujan et al., 2020).

To this end, we study the task-specific metrics on models with permuted weights (which we call **reset networks** (Zhang and Bowman, 2018; Cao, Sanh, and Rush, 2021)) and verify that the circuit recovery algorithms perform worse on these models that do not have underlying algorithms. Specifically, we create the reset network by permuting the head dimension of each layer's Q, K, V matrices, and each MLP's bias term. This disrupts the functionality of the subject model, without changing many facts about the distribution of the activations (e.g. the average magnitude at every layer). In our experiment

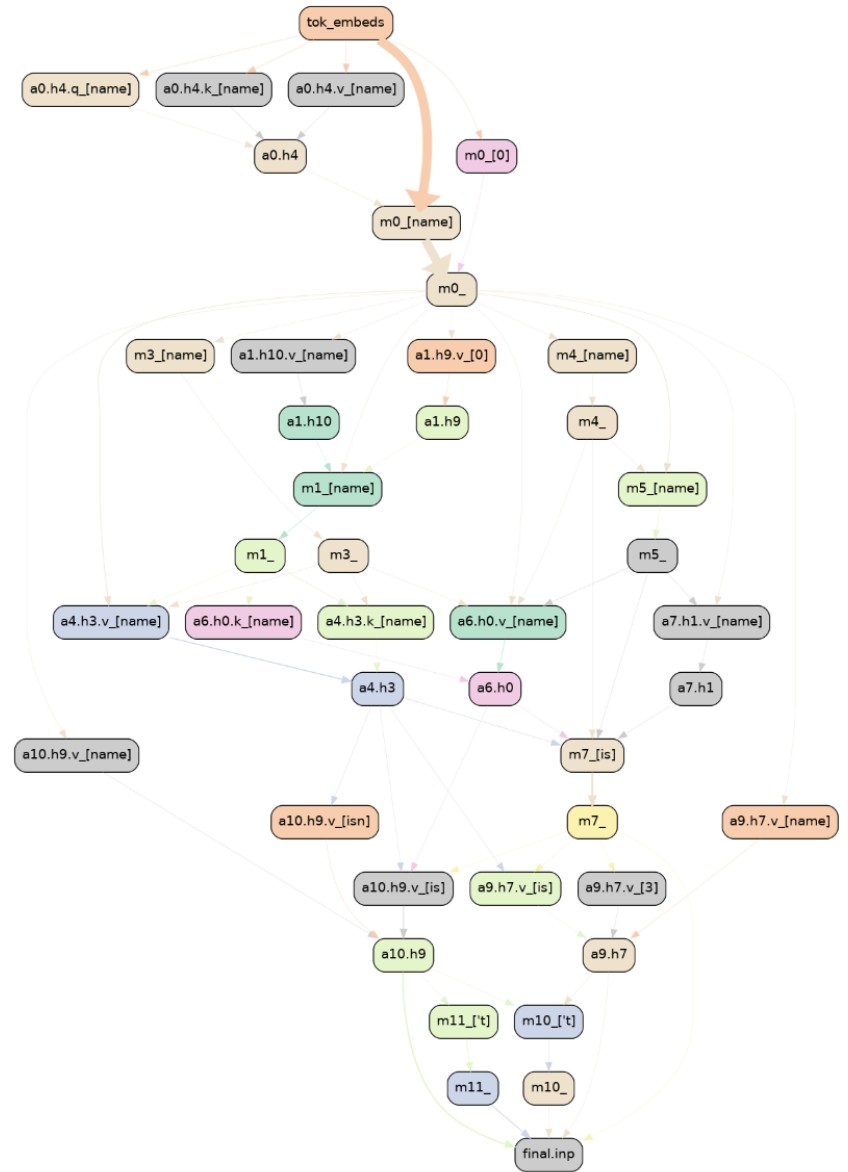

Figure 22: Gendered pronoun completion circuit found by ACDC.

in Figure 8 the metric used by each algorithm is the KL divergence between the original trained network (with no edges patched), and the activation-patched reset network.

The reset network does not exhibit the original network's behavior, and thus it should not be possible to explain the presence of the behavior. This is a strong measure of the negation of **Q2**: if the algorithm is able to find a circuit that performs the behavior on a network that does *not* exhibit the behavior, then it will likely hallucinate circuit components in normal circumstances.

# M   Automated Circuit Discovery and OR gates

In this appendix we discuss an existing limitation of the three circuit discovery methods we introduced in the main text: the methods we study do not identify both inputs to 'OR gates' inside neural networks.

OR gates can arise in Neural networks from non-linearities in models. For example, if $x, y \in \{0, 1\}$ then $1 - \text{ReLU}(1 - x - y)$ is an OR gate on the two binary inputs $x, y$.[13] To study a toy transformer model with an OR gate, we take a 1-Layer transformer model two heads per layer, ReLU activations and model dimension 1. If both heads output 1 into the residual stream, this model implements an OR gate.[14] Our dataset (Section 2) is then equal to a single prompt where the heads output 1, and we use zero activations to test whether the circuit discovery methods can find the two inputs to the OR gate.

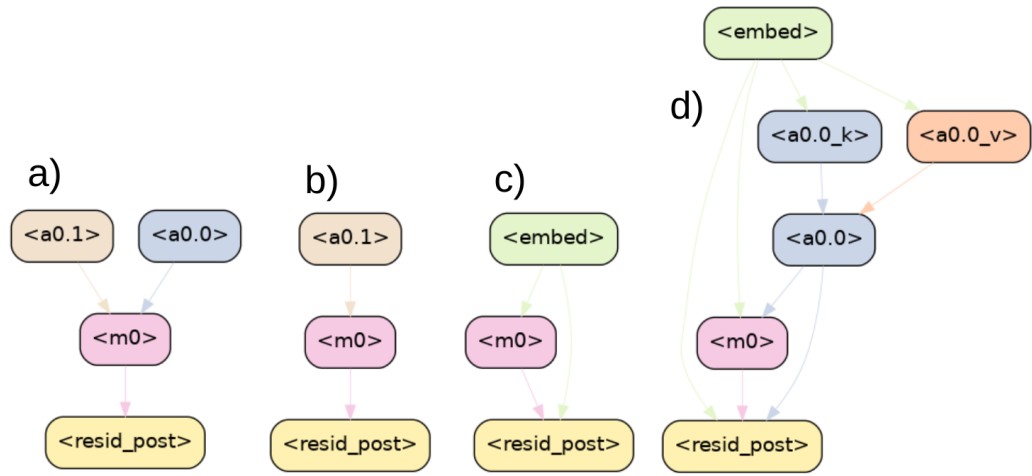

Figure 23: OR gate recovery. a) the ground truth, b) ACDC, c) HISP, d) SP.

The results can be found in Figure 23. The ground truth in a) is our toy model of an OR gate, where MLP0 performs OR on the bias terms of a0.0 and a0.1. These are the only edges that should be included. b) ACDC only recovers one OR gate input. This is because the iterative algorithm prunes the first input to the OR gate it iterates over and then keeps the other. c) HISP recovers neither OR gate input (and also recovers the unnecessary input node). d) SP recovers only one OR gate input, and several additional nodes. SP and HISP found extra edges since they include the input node by default. HISP doesn't include either attention head showing the limitations of gradients in this idealized case. We are unsure why SP finds the a0.0's key and value inputs. This shows the limitations of node-based methods for finding circuits, though ACDC is also limited. Of course, many easy fixes exist to this problem, but the priority of future work should be to explain in-the-wild language models, where it is less clear which algorithmic improvements will be most helpful. For example, follow up work found that using gradient approximations on the edges of a computational graph was very effective (Syed, Rager, and Conmy, 2023), despite not being more effective at finding OR gates.

---

[13]This construction is similar to the AND gate construction from Gurnee et al. (2023) Appendix A.12.

[14]For specific details of the TransformerLens (Nanda, 2022) implementation, see https://github.com/ArthurConmy/Automatic-Circuit-Discovery/blob/main/acdc/logic_gates/utils.py#L15.

# N    Connection to Causal Scrubbing

This work was inspired by work on Causal Scrubbing (Chan et al., 2022). The scope of each algorithm is quite different, however.

Causal Scrubbing (CaSc; Chan et al., 2022) aims primarily at hypothesis testing. It allows for detailed examination of specified model components, facilitating the validation of a preconceived hypothesis.

ACDC, on the other hand, operates at a broader scale by scanning over model components to generate hypotheses, each of which is tested using CaSc. ACDC chooses to remove an edge if according to the CaSc criterion, the new hypothesis isn't much worse.

Why is testing every hypothesis with Causal Scrubbing not incredibly inefficient? The reason is that ACDC only considers a small class of CaSc hypotheses, where paths through the model either matter, or don't matter. In effect, the CaSc hypotheses considered by ACDC don't allow any interchanges if the node "matters" (by having a unique value for each possible input), and the nodes that don't matter are each replaced by the same second data point.

Both methods currently face computational inefficiencies, albeit for different reasons and at different scales. Causal Scrubbing is impractical for somewhat complicated causal hypotheses because of *treeification*: there are exponentially many paths through a branching DAG, and each needs part of a forward pass. For ACDC, each hypothesis is quick to test, but the number of edges to search over can be quite large, so it still takes a while to search over circuits for realistic models. This problem is partially addressed by using gradient-based approcahes like Attribution Patching (Syed, Rager, and Conmy, 2023) or perhaps an edge-based version of Subnetwork Probing (Cao, Sanh, and Rush, 2021).

In summary, ACDC and Causal Scrubbing are complementary tools in the analysis pipeline. ACDC can do an initial coarse search over hypotheses and, while it is built on Causal Scrubbing, only considers a small class of hypotheses so it stays relatively efficient. In contrast, Causal Scrubbing offers a methodical way to test hypotheses, which can also specify the information represented in each node.

