# OpenReview forum: "Towards Automated Circuit Discovery for Mechanistic Interpretability"
_NeurIPS.cc/2023/Conference — NeurIPS 2023 spotlight_

### Official Review · Reviewer_aEyk · 2023-07-01

**Soundness:** 4 excellent
**Presentation:** 4 excellent
**Contribution:** 4 excellent
**Rating:** 9
**Confidence:** 4

**Summary:**

This paper first presents an overview and a useful distillation of existing mechanistic interpratility work on discovering interpretable circuits in transformer models. They say most existing work happens in 3 steps:
1. Observe a behavior (or task) that a neural network displays, then create a dataset to measure this behavior
2. Define the scope of interpretation: Do we want to look at which attention heads, mlp layers or individual neurons are important.
3. Perform a search with patching experiments to remove as many unnecessary components as possible

They then propose an algorithm ACDC to automate step 3 of this process, which has typically required extensive manual effort by researchers. They evaluate this extensively by applying it on existing circuits found by researchers, finding that it can discover existing circuits with good accuracy and outperform baselines.

**Strengths:**

- Very important/impactful problem
- Good survey of existing work in a very new topic, bringing important clarity/systemization to the workflow of these otherwise individual findings.
- Clear writing
- Extensive and comprehensive evaluation
- Good performance with the automated method
- Proposed systemization and automated method to discover circuits will likely greatly improve the speed at which new discoveries can be made in this field, making it easier for new researchers to approach.

**Weaknesses:**

- Choice of threshold parameter tau seems inconsistent/varies by task. Unclear how I would choose tau when applying this method on a new task.
- Does not address how to come up with the task and dataset for step 1 which may be the hardest part of the workflow

**Questions:**

- How did you choose tau for different experiments?

**Limitations:**

Yes, very good discussion.

---

> ### Author Rebuttal · Authors · 2023-08-10
>
> Thank you for your helpful feedback on our work! We appreciate the concise distillation of our identified three steps of the mechanistic interpretability workflow and how you highlighted the diverse strengths of our work.
>
> This response is intended to address both of the weaknesses you brought up regarding our paper.
>
> > Does not address how to come up with the task and dataset for step 1 which may be the hardest part of the workflow
>
> Overall, we don’t think that selecting tasks and datasets (Step 1 of our workflow) is likely to be as difficult or time-intensive as patching and subgraph-finding (Step 3, which we automate). For example, the IOI and Greater-Than papers used datasets consisting of at most 100 prompts that they could concisely describe, but they report much more extensive patching experiments. It is possible that in future for more challenging tasks it will be harder to design datasets, but currently we think that our contribution automates most of the identified mechanistic interpetability workflow.
>
> > Unclear how I would choose tau when applying this method on a new task
>
> Thank you for the feedback, and this is a valid concern. However, we have not found this an issue yet, particularly compared to the alternative circuit recovery methods. ACDC is an iterative rather than an end-to-end algorithm. This means that we can observe the subset of the subgraph that ACDC has recovered when only one node has been iterated over (which typically takes <1% of the total number of iterations). Therefore practitioners can use the number of recovered nodes as inputs to the output node as an approximation to the number of nodes that will be recovered in the entire ACDC process. A good example of this working is in Figure 13 in the gendered pronouns use case.
>
> We will highlight how practitioners have dealt with the $\tau$ parameter choice in an additional paragraph in the gendered pronouns appendix to describe this workflow. In the main text, we performed sweeps with logarithmic spacing between choices of $\tau$, which we will also detail in the release of results in the open source implementation.

---

> > ### Comment · Reviewer_aEyk · 2023-08-12
> >
> > Thanks for the response! This mostly addresses my concerns, and after reading the other reviews I found no significant concerns and would like to stand by my original score.

---

### Official Review · Reviewer_pJGB · 2023-07-05

**Soundness:** 3 good
**Presentation:** 2 fair
**Contribution:** 3 good
**Rating:** 7
**Confidence:** 4

**Summary:**

This paper introduces a method for pruning nodes in a computation graph that is meant to be used in the context of mechanistic interpretability, i.e., to find sub-graphs that explain/reproduce certain behavior of the overall graph while being much smaller.

**Strengths:**

- The problem is well-motivated, and the method's description is easy to understand.
- The proposed method is compared against baselines in the form of existing pruning methods that got adapted to the context of mechanistic interpretability.

**Weaknesses:**

- While the authors mention that there is early evidence that their method can produce new insights, I believe these should be highlighted more prominently. The practical relevance of this approach can be better shown by applying this method to a new setting, producing mechanistical interpretability hypotheses for some network(s), and then verifying these post-hoc.

**Questions:**

- L63: This sentence can be ambiguous for readers unfamiliar with this topic: Do the authors mean that one creates a dataset to train a new model that one can analyze?
- L93: Add a reference to explain what is meant by "tracr".
- L125ff: It can be a bit confusing that the authors here mention that they automate all but the last steps in this work, while before and after, they say they only want to automate the third step.
- L133f: What happens if the network implements a mechanism using redundant features and an or-operation? Then the detected circuit will only include one of the two valid sub-circuits (as only removing one does not impact the model much, but removing both drastically impacts it). Depending on what one is interested in investigating, this might not be an issue, but this should be discussed.
- L133f: Is this guaranteed to find the sparsest graph, or can premature pruning of connections at the end prevent the pruning of other connections that, all things considered, contain much more connections/components? For example, think about a situation where two parallel branches have different purposes but are restricted to the dataset in question, they behave identically. Both of them can be pruned away, but if the size of the branches differs, it matters which one is removed.
- Figure 4: What does this figure tell us? Adding a small description of what is shown here and a conclusion will make this more accessible to readers.
- L315f: Can the authors propose any reasonable strategy for automatically tuning/setting this hyperparameter?

**Limitations:**

The authors addressed most limitations in the main part of the paper, except that point raised in the Questions section.

---

> ### Author Rebuttal · Authors · 2023-08-10
>
> Thank you very much, pJGB, for your comments on our work! We hope that this reply answers all your questions, and look forward to further discussion.
>
> **New insights from the method should be featured more prominently.** We agree that showing the practical relevance of new methods is important evidence for their usefulness. We will add an extra sentence in the conclusion highlighting that practitioners have begun to verify the hypotheses generated by ACDC, which we have included at the end of this response. We will also add a paragraph to the gendered pronoun (Appendix I) on how the hypothesis generated thanks to ACDC was faithful to the model’s computation; also attached at the end of this response.
>
> **What does Figure 4 tell us?** We have added a clearer figure (attached in the general comment PDF) with a better caption, and will clarify in the main text. We used held-out i.i.d test data for this figure rather than the same data the circuit discovery methods used. The takeaway is that the recovered circuits get close to behaving like the full model with few edges.
>
> **Do the authors mean create a dataset and train a new model to analyze?** No, we use “dataset” to refer to prompts that show that a model has a particular behavior. We do not update the models’ weights. We will add a clarification to the main text.
>
> **"tracr" reference; automating all or one step?** Thank you for pointing out these mistakes! Tracr should have a reference on first appearance, and we only automate the third step.
>
> **Redundant features and OR-operation, ACDC will include one subcircuit.**
> This is a very good point, and it is correct. As a concrete example, consider a NN implementing an OR gate, where both inputs to it are set to the same value (both 1 or both 0). ACDC will recover exactly one of the OR gate's inputs: first it will attempt to remove one of them, and see that the circuit behavior is unchanged; then removing the second connection will impact the behavior, and it is recognized as important. SP should behave in the same way.
>
> In contrast, HISP should not recognize any of the inputs as important. The gradient of the output with respect to OR gate inputs should be zero, since both inputs are held to 0 or 1.
>
> We have conducted an experiment showing this, and put the recovered circuits in the general response PDF. We set the weights of a small ReLU transformer by hand, so it implements an OR gate. We then apply each of the algorithms, and they behave as expected: ACDC and SP find exactly one input, and HISP does not find any inputs to such an OR gate. We have included the key figure in the general comment PDF.
>
> We have written an Appendix for this experiment, which expands on the three previous paragraphs. In it, we discuss how future work on automating circuit discovery could deal with this issue. For example, ACDC-like methods could run several times, shuffling the parent nodes. Each run will recover a different input to the OR gate. The recovered circuit is then the union of what is recovered in all runs. Gradient descent methods like SP can run with different random seeds.
>
> Thank you for finding this interesting property!
>
> **Two parallel branches of different size.** This case seems similar to the OR gate example, where circuit recovery algorithms behave differently. We will include it in the OR-gate Appendix.
>
> **Sparsest graph guarantee?** None of the algorithms is guaranteed to find the sparsest graph or the branch with the
> largest effect size. Both ACDC and SP get stuck in local optima, though in practice SP seems to get stuck less often due
> to the smooth objective and gradient-based optimization. We will mention this and the previous point in the OR-gate appendix, too.
>
> **Can the authors propose any reasonable strategy for automatically tuning/setting this hyperparameter?**
> As also discussed with reviewer aEyk in more detail, in practice the iterative nature of ACDC makes selecting an appropriate parameter easier than expected. In short, the number of edges ablated early in ACDC runs can be used as a proxy for the total proportion of edges ACDC ablates. We will mention the issue in our paper by explaining more clearly how practitioners did not have trouble setting $\tau$ in real-world use cases (Figure 13 has similar sparsity at the output node to all other locations in the network). We also the open source implementation of ACDC should let the community quickly develop better strategies to tune $\tau$.
>
> ---
>
> ## Addendum: Additions To The Paper
>
> **Addition to maintext (Line 309):**
> Further, there is early evidence of the use of ACDC to help with novel interpretability work, discovering a surprising outline of a subgraph of GPT-2 Small that predicts gendered pronoun completion, where practitioners have used ACDC to generate a circuit outline of the most important pathway through a model's computation, and checked that this reflects the model's computation in normal (unablated) forward passes.
>
> **Addition to Appendix I (Line 834):**
> ACDC's output shows that the important internal information flow for predicting the expected gender has three steps. Firstly, Layer 0 attention head 0.4 and MLP0 use the name embedding which they pass (through intermediary MLPs) via key- and value-composition (Elhage et al., 2021) to attention heads 4.3 and 6.0. Secondly, heads 4.3 and 6.0 attend to the name position to compose with 0.4 and MLP0. Finally, through value-composition with attention heads 6.0 and 4.3 (via MLP7), the outputs of 10.9 and 9.7 output the expected gendered completion to the output node. Anonymous (2023) then verified that in a normal forward pass of GPT-2 Small, 0.4 has an attention pattern to itself at the name token, attention heads 4.3 and 6.0 attend to the previous name token, and 10.9 and 9.7 attend to the " is" token. They also perform path patching experiments on intermediate nodes to provide further evidence of the importance of the pathway through the " is" token.

---

> > ### Comment · Reviewer_pJGB · 2023-08-14
> >
> > Thank you for your detailed response and explanations. I'm happy to recommend accepting this paper and will increase my score accordingly.

---

### Official Review · Reviewer_Zemb · 2023-07-07

**Soundness:** 4 excellent
**Presentation:** 4 excellent
**Contribution:** 3 good
**Rating:** 7
**Confidence:** 5

**Summary:**

This paper proposes an approach for the automatic discovery of circuits (ACDC) in artificial neural networks (applied to transformer-based LLMs), which works by recursively constructing a subgraph of "important" nodes identified through the patching of model activations on datapoints relevant to a specific task (the choice of which is, in general, non-trivial). The authors demonstrate, through many experiments, that ACDC is mostly able to faithfully recover circuits which were manually identified by previous researchers on a variety of tasks (notably Python docstrings, IOI and induction heads), thus automating a highly labor-intensive part of the circuit discovery process. Additionally, they explore choices of patching value, target metric and threshold value, whilst also demonstrating that other comparable methods for distillation/subgraph isolation are not as well-behaved as ACDC.

In addition to the paper, the authors also release an open-source implementation of ACDC which has already been applied to some success by other mechanistic interpretability researchers already.

**Strengths:**

The methodology for automatic circuit discovery proposed by the paper extends previous approaches for activation patching to automate otherwise labor-intensive mechanistic interpretability work. This in of itself is not a significant novelty, but the paper's strength lies in a thorough experimental investigation of the benefits of ACDC over other subgraph discovery methods, coupled with new methodological insights on how best to perform activation patching. In particular, they provide two novel findings: 1) KL Divergence is more well-behaved than logit differences when performing activation patching for circuit discovery, and 2) Zero patching, whilst significantly OOD, is often more effective than patching corrupted activations.

The presentation of the paper is very good, with a coherent narrative for the experimental investigations and clear figures supporting all claims. Additionally, the supplementary materials provide further interesting discussion and results, which given the exploratory nature of circuit discovery is highly valuable.

Lastly, the release of the accompanying ACDC algorithm for use by the community is a significant contribution in of itself, as demonstrated by the fact that other members of the MechInt community have already applied ACDC for their own research.

**Weaknesses:**

No major weaknesses were identified.

There are a fair number of minor phrasing issues outlined in the following nitpicks section. In the related work section, explicitly stating how path patching varies procedurally from ACDC might be worthwhile. Additionally, a discussion of how ACDC varies from Causal Scrubbing in its patching methodology may be useful.

## Nitpicks
* 3: makes it **too** costly"
* 27: circuits **as** subgraphs
* 32: "with which to extract" or "for extracting"
* 32: remove "that automates part of it"
* 85: The choice of phrasing - "clearly defined behaviour" makes this sentence almost tautological. Perhaps an explicit mention of simplicity would be suitable here. Researchers unfamiliar with MechInt may consider e.g. "Writing python code" clearly defined or "writing python docstrings" too broadly defined.
* 91: "**Tasks** 1 and 3" inside the parentheses
* 94: from **each task, which** researchers
* 98: **as** a computational graph
* 101: "on the level of detail of _their_ explanations of model behaviour* subject unclear and wording confusing
* 125: such as -> for example (as subject of "such as" could be "tasks"). "predict correct gender predictions" remove last predictions?
* 223: It seems the discussion of zero ablations takes place in Section 5 and Appendices F.2, rather than Appendix D.
* 233: "we explain how compare to" -> "we compare to"
* 236: "experiments use the same modifications to SP and HISP *as*"
* 308 "is known*, and through comparison with previous*..."
* 310: not clear what "outline of a subgraph" means vs. just "a subgraph"
* 318: "work; a novel contribution" or "work - a novel contribution".
* 320: "*within* the community"




**Questions:**

The following questions are not crucial to the narrative of the paper, nor potential critiques of completeness.

1. Have you thought of automated ways to trade-off circuit faithfulness versus sparsity, without re-running ACDC with different values of $\tau$?
2. In general, how sensitive is ACDC to the choice of clean and corrupted datapoints. E.g. for the IOI task does ACDC provide considerably different circuits if very few examples are provided, vs. many? What about paraphrasing or noise injection as in ROME (this is probably very task dependent)?
3. When modifying Subnetwork Pruning you discuss interpolating the mask values - should we expect that linearly interpolating between a clean and corrupted activation is principled (i.e. does not potentially shift the representation to something meaningful but distinct)?
4. Are all values of SP masks 0, or 1 by the end of SP training? If not, then for the sake of counting subgraph edges, what is considered an "unmasked node"?
5. Given the unexpected utility of using zero ablations, would the authors suggest trying this whenever utilizing ACDC?
6. Could the authors expand on how the _locally significant changes_ alternative to detecting salient parts of the subgraph would avoid potential sensitivity to the form of patching? Presumably, some perturbation would be required to measure "effects" and it is not immediately clear what this perturbation is, if not an activation patch.


**Limitations:**

Limitations are addressed, or explored through supplemental experiments.

---

> ### Author Rebuttal · Authors · 2023-08-10
>
> We thank the reviewer for their kind words about our work. In particular, we are happy to see that they are also excited about open-sourcing ACDC and that they recognize how it has already been used by the community to accelerate mechanistic interpretability research. We were also pleased to see that you (reviewer Zemb) were complimentary of our empirical evaluations on KL divergence and zero patching. We hope our replies to your questions are comprehensive. We have added the nitpicks to our working copy and thank you for documenting them.
>
> Regarding the call for more description of how ACDC differs from causal scrubbing we will add the following paragraph to an appendix, but in summary: we think that **ACDC and Causal Scrubbing are complementary tools—Causal Scrubbing is a tool to test hypotheses, while ACDC focuses on the generation of good hypothesis**
>
> Addition to appendices: “Both Causal Scrubbing (Chan et al. 2022) and ACDC make extensive use of path patching: Both test how well a circuit reproduces a model's performance by resample-ablating all edges other than the ones specified by the circuit. This path-patching test is a special case of Causal Scrubbing.
>
> The step automated by ACDC is the generation of good hypotheses, it allows us to find the smallest circuit hypotheses that reproduce certain levels of model performance in an efficient and automatic manner.
>
> Causal Scrubbing itself is not limited to testing just model subgraphs (as we do here) but in principle also allows testing which part of the inputs are relevant for which parts of the circuit, while ACDC automates only the circuit finding on a subgraph level.”
>
> Now we will respond to your questions.
>
> > Have you thought of automated ways to trade-off circuit faithfulness versus sparsity, without re-running ACDC with different values of $\tau$?
>
> This is an interesting point. Like many ML algorithms, the performance of ACDC is somewhat sensitive to the value of tau. In general, we think that it is not too much of a burden to do a sweep over tau values—especially compared to the previous state-of-the-art of manually searching for circuits by hand. In developing our work, we found that the number of connections pruned so far is highly predictive of how many connections will remain at the end of an ACDC run. We provide evidence for this claim in Appendix I, and will add a paragraph to this Appendix in response to the reviewers' interests.
>
> > In general, how sensitive is ACDC to the choice of clean and corrupted datapoints … if very few examples are provided, vs. many?
>
> The datasets used for the behaviors we investigate are very small. For example, we 40 dataset examples are used for the induction task and 100 for IOI and Greater Than. Because we are able to get good results on these small datasets, we do not expect the choice of data points to significantly affect ACDC’s output. In the attached PDF (Figure 4) we evaluated the KL Divergence on held-out test examples, but we have not extensively evaluated this. We did not compare noisy corruptions because we in general we want to test the ability to isolate specific behaviors (e.g the IOI and Greater Than circuits carefully choose patching distributions so that they can isolate a specific behavior present in one distribution, but not in the other). We included zero ablation comparisons as this intervention does not require any parameter choice (but choosing a norm term for noise interventions is an additional choice).
>
> > Are all values of SP masks 0, or 1 by the end of SP training? If not, then for the sake of counting subgraph edges, what is considered an "unmasked node"?
>
> We round the outputs of SP to 0 or 1 and then count edges with the rounded graph.
>
> When modifying Subnetwork Pruning you discuss interpolating the mask values - should we expect that linearly interpolating between a clean and corrupted activation is principled (i.e. does not potentially shift the representation to something meaningful but distinct)?
>
> We use linear interpolation as a continuous approximation through training that uses resampled activations (as in Causal Scrubbing). The generally good performance of SP makes us confident that our continuous approximation that we then clamp to 0 or 1 is a valid use of the SP approach. Nevertheless, we take your point that this could potentially shift the representation somewhere else that is in distribution but distinct (i.e. meaningful but distinct). We think further developments on our approximation is an interesting avenue for future work (adapting gradient based methods to work with corrupted activations).
>
> > Given the unexpected utility of using zero ablations, would the authors suggest trying this whenever utilizing ACDC?
>
> We apologize to the reviewer for the data in our incorrect appendix figure (Figure 15, see our global response and the attached PDF) that may have led to this conclusion. We attach in the PDF the correct performance of ACDC, SP and HISP with zero activations on the IOI task in Figure 15, and find that zero ablation performs worse for this task (like Docstring and Greater-Than). ACDC performance of zero ablations on the tracr tasks was unchanged (perfect) but we do not think that this should be extrapolated to realistic language models.
>
> > Could the authors expand on how the locally significant changes alternative to detecting salient parts of the subgraph would avoid potential sensitivity to the form of patching?
>
> We don't understand how the locally significant alternative “would avoid potential sensitivity to the form of patching”. In our paper, we discussed how the locally large effects alternative could resolve some issues with negative head recovery, although since it is not optimizing any metric globally it would be more difficult to interpret the performance of this alternative, unlike with e.g KL Divergence.

---

> > ### Comment · Reviewer_Zemb · 2023-08-14
> >
> > Dear Authors,
> >
> > Thank you for your comprehensive response to my questions.
> >
> > * The provided summary of Causal Scrubbing vs ACDC is very clear, and appreciated
> > * The discussion of the sensitivity with respect to the datapoint choices is convincing and I don't think further evaluation is necessary. The relative stability of the results is reassuring
> > * As linear interpolation on SP does indeed yield good results, your confidence in its validity does seem well-placed, as it is difficult to imagine how undesirable "semantically meaningful" interpolations (which I had worried about) wouldn't manifest without also decreasing performance
> > * It is good to know that zero ablations aren't as effective as I had mistakenly concluded!
> > * This is clear now - it seems I had some initial confusion about "metric" vs. patching when reading this section, though It seems quite clear
> >
> > Given the scope of ACDC's application, I retain the current rating and recommend this paper for acceptance as a sound and meaningful contribution to the field of Mechansitic Interpretability.

---

### Official Review · Reviewer_6BEj · 2023-07-11

**Soundness:** 3 good
**Presentation:** 3 good
**Contribution:** 3 good
**Rating:** 6
**Confidence:** 3

**Summary:**

The paper is a fresh take on mechanistic interpretability, focusing on the automation of the interpretability task, demonstrating it on attention based models. The method, ACDC, finds the pareto optimal subgraphs of the network, thus bringing down the number of connections to highlight the role each unit plays in the predictions of the model. The authors present extensive experiments and analysis on various tasks, including the interesting IOI and Greater-Than, with further insight on the performance trends in the supplementary work. On providing an idea of the workflow in the domain, they introduce their algorithm focusing on iteratively chipping away at the computational graph, creating a sparse graph while retaining the good performance on the specified task and metric. They do mention the limiting prowess of the algorithm, that is the handicap of not identifying all the abstract units and sensitivity to hyper-params, while also not being automated end to end for a complete interpretability framework setup. Overall the algorithm is a good starting point to build on automating scalable interpretability.

**Strengths:**

- Neat presentation, with well organized sections and appropriate background information.
- Clean and crisp algorithm, that is easy to understand and yet highly effective in its job.
- Experiments are well defined, tasks, metrics and objectives are comprehensively made clear.
- Comprehensive ablations and detailed supplementary work to support the claims made and make sense of the findings, special kudos for the highly effective plots and task-wise analysis.
- Frank and clear understanding of the shortcomings and strengths of the algorithm make this method a clear one to build on top of, prompting further interesting work in a highly important domain.

P.S. A playful modification to the title could be to make it AC⚡DC to make it a play on the famous music band.

**Weaknesses:**

- Distinction from previous work is not very clear. Especially as compared to HISP, the top k heads are analogous to retaining only the influential heads in ACDC. While I agree that the algorithm in itself is distinct, the inspiration from previous works should be made clear. I would suggest adding a subsection to compare the circuits derived from the two methods for further comparison in the approaches (akin to Figure 6), and explicitly highlighting the specific fail cases for HISP that ACDC works for could be a great insight.
- While section 5 is helpful in answering questions about the effectiveness of ACDC, Line 253 as pointed out by the authors raises questions. In my opinion more work in establishing KLD as a faithful proxy must be undertaken.
- While Appendix K and H are helpful to establish the “correctness” of the subgraph recovered, unless one knows the true minimal graph, it is difficult to compare and decide which of the methods is “more correct”.

**Questions:**

- See weaknesses.
- While Figure 16 (Typo in the Greater Than graph with a $\mu$ perhaps?) and 17 differ significantly on the performance on the reset network and the trained network, for the SP and HISP networks, the ACDC points are invisible in Figure 16 subplots for Tracr (Reverse), Tracr (Proportion), Greater Than, I’m assuming the scale of the graph points to them being in this range, but perhaps consider replotting with the ACDC method points being visible.
- Line 313: As the authors correctly point out, the issue of missing certain units must be investigated and probed further. Highlighting taskwise the missing sections and what they could correspond to could help draw patterns on the way ACDC works and alleviate weaknesses in further work to build on top of this.
- Likewise the final hyperparameters used and ablations on the variance in the performance as one tweaks these hyperparams could be useful to note the sensitivity and brittleness of the algorithm. For a $\delta$ change in a hyperparam how does the method performance vary?

I am open to increasing the score on further clarifying discussion with fellow reviewers and authors.

---

> ### Author Rebuttal · Authors · 2023-08-10
>
> Thank you very much for your well-considered review! We are very happy that you appreciate the work we put into exploring the shortcomings and advantages of each algorithm. Thank you also for the insightful questions, and please let us know if you have any more.
>
> **Comparison to HISP.** Thank you for bringing this up! HISP is a heuristic pruning algorithm. ACDC can be seen as a pruning algorithm too. However, its goal is very different from most of the pruning literature, which we mention in related work. For example, ACDC operates on connections between heads (edges) instead of heads (nodes), which sometimes makes the forward pass run more slowly and require more memory, a very undesirable thing in NN pruning. However, as an advantage ACDC can recover the path-specific effects of edges – in Figure 6, as you mention, ACDC recovers the effect that Layer 0 heads have on the output indirectly: through their input connections to Layer 1. Both other algorithms we compared to cannot be this specific. However, further emphasizing the conceptual similarities and differences is worthwhile and we will do this in Section 3 and Appendix H.
>
> **Faithfulness of KL divergence.** As far as we understand, your concern about the faithfulness of KL in Section 4 of our work asks two questions:
> - (1) Is KL a good proxy for the task-specific metric when finding circuits?
> - (2) Does optimizing KL divergence reliably yield the circuit that’s implementing the behavior in the NN? For example, optimizing KL has to yield a circuit that contains both ‘positive’ and ‘negative’ contributions to the log-likelihood.
>
> We do not assume either of these things in the paper, and we do examine them empirically.
>
> We believe we have pretty firmly established (1) for the tasks we tried. Sadly we did not have space to include this in the response PDF, but we have plots that measure *task-specific loss on held-out test data*, on experimental runs that use the *KL as a target*. The result looks pretty much like Figures 16-19 and the updated Figure 4: the held-out task-specific loss improves monotonically with higher numbers of edges. Modifying the $\tau,\lambda$,% parameter monotonically decreases loss and increases number of edges. Thus KL is a good proxy for task-specific loss, and is not overfitting the subgraph.
>
> We have not established (2) very well, but it was a major focus of the experiments of this paper. We are bottlenecked by the lack of good measurements of hypothesis correctness in the field of interpretability, which is exactly your third objection: mostly we don’t know the true minimal graph! We attempted to establish (2) with the ROC plots (e.g. the updated Fig. 3) and the reset network experiments (Figs. 16-19). The ROCs depend on the correctness of previous work, and the other plots are weak evidence, so we still don’t know whether (2) is true. We will highlight this problem further when discussing the metrics at the beginning of Section 4.
>
> That said, now that there is a clear way to automate mechanistic interpretability, we expect ACDC to be replaced by faster and better algorithms soon. KL divergence may turn out to be a mediocre target for finding circuits! We will also emphasize this point in the conclusion.
>
> **We don’t know the true minimal graph**. Correct, sadly we don’t, and it’s a problem for your previous objection as well.
> Measuring the correctness of a hypothesis is an open problem in the field. There are some attempts in the literature (Geiger et al. 2021, Chan et al. 2022), but their soundness is only supported by theoretical arguments. Little experimentation targeted to establishing that these approaches reliably tell good from bad hypotheses has been done, instead they have just been applied.
>
> **Sensitivity of the hyperparameters.** Very good point. The best way to communicate non-linear sensitivity is via plots that show the variation in performance for each value of the ACDC ($\tau$), SP ($\lambda$) and HISP (% heads) parameters. We did so by color-coding points in Figs. 3 and 4 of the updated PDF. The number of recovered edges and the task-specific loss vary smoothly with these parameters, with little noise. Thus, this does not seem to be a problem. The main robustness challenge of these algorithms is that the dataset, metric and type of activation patching, can greatly change the recovered circuit.
>
> **Typos in graph**. The $\mu$ is not a typo, it indicates the SI prefix ‘micro’ ($\cdot 10^{-6}$). The values displayed in that plot are extremely small, practically zero. We agree this is confusing and apologize, we will amend the paper to explain this. The ACDC points are not out of range, but our data collection script had a bug and missed this particular run. We have fixed this oversight already in our copy of the paper!
>
> **Highlighting the units that the circuit misses**. The IOI negative heads are the only pattern that we noticed. After a more thorough look, we found that they are recovered by ACDC with the KL divergence as a target, when the threshold is $\tau \le 0.00398$. This is towards the mid-range of the thresholds we used in the final sweeps in the updated Figure 3, which is from $10^{-5}$ to $10^0$ (the figure goes down to $10^{-9}$, but that is only for the tracr tasks).
> Unsurprisingly, ACDC does not manage to recover the negative heads when using the IOI metric (logit difference) as a target. This is a reason to prefer the KL over the task-specific metric.

---

> ### Author Response · Authors · 2023-08-18
>
> Dear 6BEj, we hope our responses were clear. Would you like to ask any further questions?

---

### Author Rebuttal · Authors · 2023-08-10

We thank all the reviewers for their detailed and positive feedback on our paper.

*All* reviewers recommended an acceptance and together found our contribution “well-motivated” (pJGB) and a “fresh take on mechanistic interpretability” (6BEj) that is an “important clarity/systemization to the workflow of these otherwise individual findings” (aEyk) and automates “a highly labor-intensive part of the circuit discovery process” (Zemb).

We’ve attached a PDF with updated figures in this global response, and in our individual responses to reviewers we hope that we have answered specific questions.

Unfortunately, we found that our code for the ROC figure was buggy in a subtle way. Our conclusions from the main text remain the same (e.g ACDC has the greatest performance by AUC), though the updated figure in the PDF does improve the performance of the existing algorithms that we repurposed for mechanistic interpretability. One minor point in the appendix was affected, which we discuss with reviewer Zemb.

Our PDF includes
1. An updated ROC plot (Figure 3) with fixed bugs from the earlier figure.
2. An update to the Pareto frontier plot (Figure 4) for clarity (there were no problems with the data collection for this plot) as pointed out by reviewer pJGB.
3. An update to the appendix Figure 15, which shows that zero ablation does not work as well in that case.
4. A figure with experimental results on automated circuit discovery of OR gate mechanisms, to respond to reviewer pJGB. We thank the reviewer for showing interest in this issue with circuit discovery, and we will add this example as an appendix to our paper.

We look forward to engaging with the reviewers further on their questions. Thank you all for your work!

---

### Decision · Program_Chairs · 2023-09-21

**Decision:**

Accept (spotlight)

**Comment:**

This paper proposes an algorithm for a subtask frequently performed in mechanistic interpretability: automatic discovery of circuits. The algorithm (ACDC) recursively constructs a subgraph identified through the patching of model activations on datapoints relevant to a specific task. The authors demonstrate that ACDC is mostly able to faithfully recover circuits which were manually identified by previous researchers on a variety of tasks (notably Python docstrings, IOI and induction heads). They also demonstrate other comparable methods for distillation/subgraph isolation are not as well-behaved as ACDC.

Overall, mechanistic interpretability currently relies on a lot of tailor-made routines, arrived through laborious trial-and-error and task knowledge. Methods which successfully automate any part of the pipeline are much welcome, and will generate interest at the conference.